# Plasmid-mediated phenotypic noise leads to transient antibiotic resistance in bacteria

J. Carlos R. Hernandez-Beltran [1,2] ✉, Jerónimo Rodríguez-Beltrán [3], Oscar Bruno Aguilar-Luviano[1], Jesús Velez-Santiago[1], Octavio Mondragón-Palomino [4], R. Craig MacLean [5], Ayari Fuentes-Hernández[1], Alvaro San Millán [6] & Rafael Peña-Miller [1] ✉

The rise of antibiotic resistance is a critical public health concern, requiring an understanding of mechanisms that enable bacteria to tolerate antimicrobial agents. Bacteria use diverse strategies, including the amplification of drug-resistance genes. In this paper, we showed that multicopy plasmids, often carrying antibiotic resistance genes in clinical bacteria, can rapidly amplify genes, leading to plasmid-mediated phenotypic noise and transient antibiotic resistance. By combining stochastic simulations of a computational model with high-throughput single-cell measurements of $bla_{TEM-1}$ expression in *Escherichia coli* MG1655, we showed that plasmid copy number variability stably maintains populations composed of cells with both low and high plasmid copy numbers. This diversity in plasmid copy number enhances the probability of bacterial survival in the presence of antibiotics, while also rapidly reducing the burden of carrying multiple plasmids in drug-free environments. Our results further support the tenet that multicopy plasmids not only act as vehicles for the horizontal transfer of genetic information between cells but also as drivers of bacterial adaptation, enabling rapid modulation of gene copy numbers. Understanding the role of multicopy plasmids in antibiotic resistance is critical, and our study provides insights into how bacteria can transiently survive lethal concentrations of antibiotics.

The evolution and spread of antimicrobial resistance in clinical pathogens represent a major public health problem that threatens to become a global crisis[1]. Drug resistance is traditionally thought of as a stable trait mediated by mutations in existing genes or by the acquisition of dedicated drug resistance genes[2]. However, it has become increasingly clear that resistance can also arise from a small fraction of cells that are tolerant to antibiotics[3] and can lead to treatment failure[4,5].

Multiple genetic and metabolic mechanisms generate subpopulations with varying levels of drug resistance. For example, some bacterial populations harbor a subset of dormant cells, known as persisters, that survive drug exposure and resume growth once the antibiotic is withdrawn[6]. Other tolerance mechanisms are based on the heterogeneous production of drug-degrading enzymes[7,8] or signaling molecules[9], which generate subpopulations with different degrees of drug susceptibility. Additionally, bacterial populations may exhibit stochastic expression of genes encoding intrinsic antibiotic resistance mechanisms, notably efflux pumps[10,11].

[1]Center for Genomic Sciences, Universidad Nacional Autónoma de México, 62210 Cuernavaca, México. [2]Department of Microbial Population Biology, Max Planck Institute for Evolutionary Biology, 24306 Plön, Germany. [3]Department of Microbiology, Ramón y Cajal University Hospital (IRYCIS) and CIBERINFEC, Madrid, Spain. [4]Laboratory of Parasitic Diseases, National Institute for Allergy and Infectious Diseases, National Institutes of Health, Bethesda, MD 20892, USA. [5]Department of Biology, University of Oxford, OX1 3SZ Oxford, UK. [6]Department of Microbial Biotechnology, Centro Nacional de Biotecnología - CSIC, 28049 Madrid, Spain. ✉e-mail: hernandezbeltran@evolbio.mpg.de; rpm@ccg.unam.mx

Phenotypic heterogeneity in isogenic bacterial populations can arise from variations in protein levels over time and differences in protein abundance between individual cells[12]. This cell-to-cell heterogeneity can result from a range of factors, including promoter noise[13]; protein aggregates[14]; asymmetry in the cell division process[15]; or stochastic fluctuations in the concentrations of proteins, mRNAs, and other macromolecules present at low-copy numbers in the cell[16–18]. Our study proposes a novel mechanism for producing phenotypic noise that is mediated by the stochastic nature of plasmid population dynamics.

Plasmids are DNA molecules that replicate independently of the chromosome and can carry resistance genes, with some present in multiple copies per cell[19]. A recent clinical study showed that a large fraction of pathogenic *Escherichia coli* isolates carry small ColE1 plasmids[20]. The number of plasmids carried by each cell is a key driver of virulence[21] and horizontal gene transfer[22]. The interaction between replication and segregation, and the complex population dynamics this produces[23,24] is known to enhance bacterial adaptation to novel environmental conditions[25], as well as to determine the repertoire of genes carried in plasmids[26] and their stability in the absence of selection[27,28].

Previous studies have shown that exhibiting phenotypic heterogeneity can be a microbial effective adaptive strategy to survive fluctuating environmental conditions[29–31]. In this study, we combine computer simulations with single-cell and population-level experiments to show that heterogeneity in plasmid copy number generates phenotypic variability that can have significant implications for the survival of bacterial populations exposed to a $\beta$-lactam antibiotic. We conclude by arguing that encoding drug-resistance genes in multicopy plasmids is beneficial for bacterial populations in rapidly changing environments, by enabling a reversible phenotypic resistance mechanism based on the stable coexistence of cells with varying plasmid copy numbers.

## Results

### Stochastic plasmid dynamics promotes adaptation to fluctuating selection

To explore the interaction between stochastic plasmid dynamics and fluctuating selection for plasmid-encoded genes, first, we used a multi-level computational model that incorporates intracellular plasmid dynamics into an ecological framework (Supplementary Information). Briefly, this agent-based model explicitly simulates key cellular processes, such as random plasmid replication and segregation, resource-dependent growth, cell duplication and antimicrobial-induced death. The propensities of growth and death are determined from the concentrations of a limiting resource and a bactericidal antibiotic present in a well-mixed environment.

Figure 1A shows numerical realizations of the model simulating an exponentially-growing population of cells descended from a parental plasmid-bearing cell. We considered the number of plasmids carried by each cell as a time-dependent variable subject to two main sources of noise: (1) imperfect plasmid copy number control[32], with plasmid replication occurring in discrete events distributed stochastically over time, and (2) plasmid segregation occurring randomly between daughter cells upon division[33]. A consequence of this stochastic plasmid dynamics is that PCN of individual cells is highly variable over time (Fig. 1D). This cell-to-cell heterogeneity results in a PCN distribution with large variance (Fig. 1B).

By assuming a linear relationship between PCN and the expression level of plasmid-encoded genes, we estimated the probability of an individual cell dying upon exposure to a given antibiotic concentration from the number of plasmid copies it carries and the degree of resistance conferred by each plasmid-encoded gene. For instance, if we assume that every cell in the population is equally sensitive to the antibiotic (i.e., a population with a low-variance PCN distribution), then

there exists a drug concentration that kills all cells simultaneously (a dose referred to in the clinical literature as the minimum inhibitory concentration, MIC). Hence, the survival probability function of such a homogeneous population is a stepwise function that switches from 1 to 0 at this critical drug concentration (black dotted lines in Fig. 1B, C). However, when we consider a heterogeneous population characterized by a PCN distribution with large variance, the population contains cells with fewer or more gene copies than the expected value (green lines in Fig. 1B). This implies that the survival probability of heterogeneous populations is lower than that predicted for a homogeneous population at sub-MIC concentrations and higher than the predicted value in high-drug environments (Fig. 1C).

In our computational model, we observed that exposure to antibiotics led to a reduction in total bacterial density. Crucially, as cells with lower levels of resistance were eliminated first from the population, the remaining cells had higher plasmid copy numbers, leading to a shift in the PCN distribution towards higher values (red lines in Fig. 1D). We also found that the degree of drug-induced PCN amplification was directly proportional to the strength of the selective pressure (Fig. 1E). Notably, selection for high-copy plasmid cells occurred even at sub-lethal drug concentrations, as the antibiotic eliminated cells with fewer plasmids than the mean PCN. High-PCN cells that survived the antibiotic exposure resumed growth and division, leading to stochastic replication and segregation of plasmids, thus restoring the whole PCN distribution. As a result, the surviving cells rapidly produced lower-PCN cells, which had a competitive advantage over high-PCN subpopulations, resulting in the mean PCN of the population returning to pre-exposure levels. We repeated the computational experiment for different selective pressures and found that the strength of selection was not only correlated with the extent of PCN amplification but, notably, with a reversal of selective advantage favoring lower PCN after drug removal, as shown in Fig. 1F.

### Experimental modulation of PCN distributions in bacterial populations

To evaluate in vitro how the environment modifies the distribution of plasmids in bacterial populations, we used an experimental model system consisting of E. coli MG1655 carrying pBGT, a ColE1-like plasmid containing a GFP fluorescent marker (*GFPmut2*) and $bla_{TEM-1}$, a gene that encodes a TEM-1 $\beta$-lactamase, which inactivates $\beta$-lactam antibiotics by hydrolyzing the $\beta$-lactam ring[34]. $\beta$-lactam resistance genes are generally located on plasmids and, in particular, TEM-1 has a plasmid origin, with more than two-hundred TEM $\beta$-lactamase variants descending from this allele recorded[35]. We denote the strain carrying this well-characterized[36,37], non-conjugative, and multicopy plasmid as MG/pBGT (average copy number = 19.12, s.d. = 1.53; Fig. 2A, B)[36].

As a control, we used a fluorescently tagged strain carrying a chromosomally encoded $bla_{TEM-1}$, which we term MG:GT. To explore the correlation between PCN and fluorescence, we also used strains obtained in a prior experimental evolution study[36]. These strains contain mutations in the origin of replication that lead to a higher average PCN (Table S1), resulting in elevated fluorescence intensity and increased drug resistance compared to MG/pBGT (Fig. 2D).

In a recent study, direct, fluorescent-reporter-based measurement of PCN, promoter activity, and protein abundance at single-cell resolution revealed a positive correlation between PCN and protein expression[38]. In our experimental system, we similarly observed an association between PCN measured by qPCR[25,39] and fluorescence intensity quantified using a fluorescence spectrophotometer ($R^2 = 0.9387$). To validate the correlation between PCN and GFP in our system, we sorted a population of MG/pBGT cells according to GFP intensity into clusters with low, medium, and high fluorescence and confirmed the positive association between fluorescence and mean PCN estimated by qPCR (Fig. S1).

To measure the effect of the strength of antibiotic selection on the distribution of PCN, we exposed MG/pBGT cells to a range of ampicillin

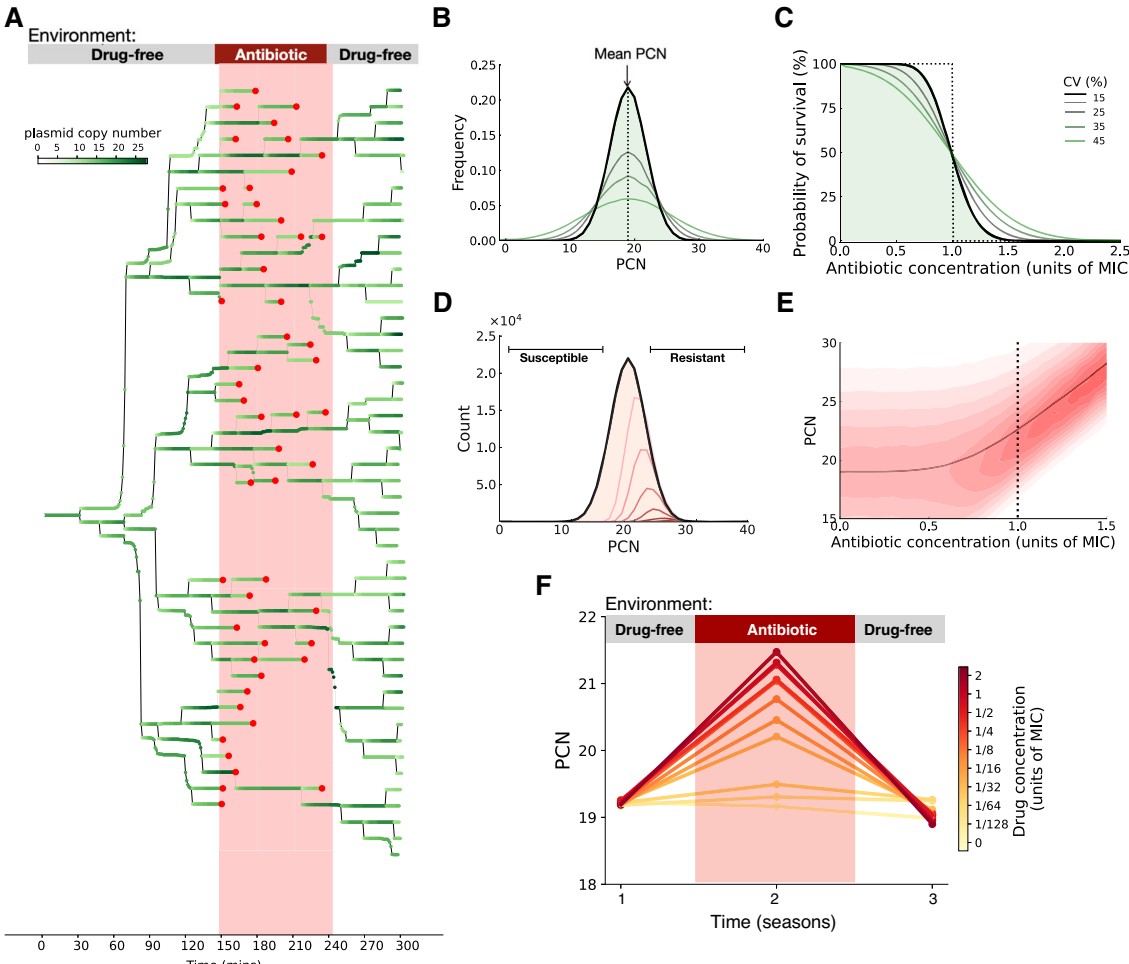

**Fig. 1 | Stochastic plasmid dynamics yield heterogeneous populations.**
**A** Simulations of a plasmid dynamics model of a population growing in a drug-free medium from a single plasmid-bearing cell. The plasmid copy number (PCN) over time is depicted using a gradient of greens. The red area indicates the time interval during which the population was exposed to a lethal drug concentration. Most cells were killed during drug exposure (death events denoted with red circles), but a small fraction of cells carrying a high PCN survived and proliferated once the antibiotic was removed. **B** Density distribution of end-point PCNs (maximum) estimated using the computational model. The black line represents the PCN distribution obtained using the parameter values described in Table S2, while the green lines depict other simulations resulting in distributions with larger variances. **C** Probability density functions of Normal distributions with a fixed mean and increasing standard deviations. The dotted line corresponds to the case when the PCN distribution has zero variance. As the variance of the PCN distribution increases, the fraction of cells with an increased probability of survival at high drug concentrations also increases. **D** PCN distributions obtained for different antibiotic concentrations: the black line represents the drug-free environment, and the distributions obtained after exposing the heterogeneous population to increasing drug concentrations are depicted in a gradient of red. **E** Mean PCN (solid line) of the PCN distributions obtained after exposing the population to a range of antibiotic concentrations. The red area corresponds to the frequency of PCN in the population for each drug concentration. The dotted line illustrates the minimum inhibitory concentration (MIC) of the homogeneous population. Note that selection for cells with multiple plasmids occurs even at sub-MIC concentrations. **F** Mean PCN at the end of each season in experiments performed with increasing concentrations of antibiotics (low doses in yellow and a lethal dose in red). Note how the increase in mean PCN observed during the selective phase of the experiment is proportional to the drug concentration. The antibiotic was removed in season 3, and the mean PCN exhibited by the population is restored to levels displayed before drug exposure.

(AMP) concentrations and used flow cytometry to measure GFP abundance of individual cells. As predicted by the model, we found that the mean GFP abundance in the population increased with the strength of selection (Fig. 2C). Concurrently, we observed a diminishing coefficient of variation (CoV) for the PCN distribution as the drug concentration increased. This decrease in CoV suggests that antibiotic selection is not merely amplifying the prevalence of plasmids within the population but is also homogenizing the PCN distribution. Under strong selective pressures, cells with insufficient plasmid numbers are selectively disadvantaged, leading to a more uniform PCN distribution among the surviving population (Fig. 2E). We also observed a rapid increase in GFP for the high-copy plasmid strains in the presence of antibiotics, a result confirmed using qPCR to estimate changes in mean PCN in the population before and after drug

exposure (Fig. S2). When the same experiment was repeated with MG:GT cells, mean fluorescence and the coefficient of variation remained constant across all AMP concentrations (Figs. S3 and 2E).

A comparison of growth rate in strains with different PCNs with respect to plasmid-free cells revealed a negative association between maximum growth rate and mean PCN in the absence of selection for plasmid-encoded genes (Fig. S13). The cost associated with bearing plasmids is well-documented[40–42], particularly for ColE1-like plasmids[43,44], and has been reported for multiple plasmid-host associations in a wide range of bacterial species[36,45–47].

The burden associated with plasmid carriage is highly variable and depends on the interaction between plasmids and their bacterial hosts[48]. This fitness cost can be ameliorated through mutations in genes located either on the chromosome or the plasmid[49–52]. In addition to

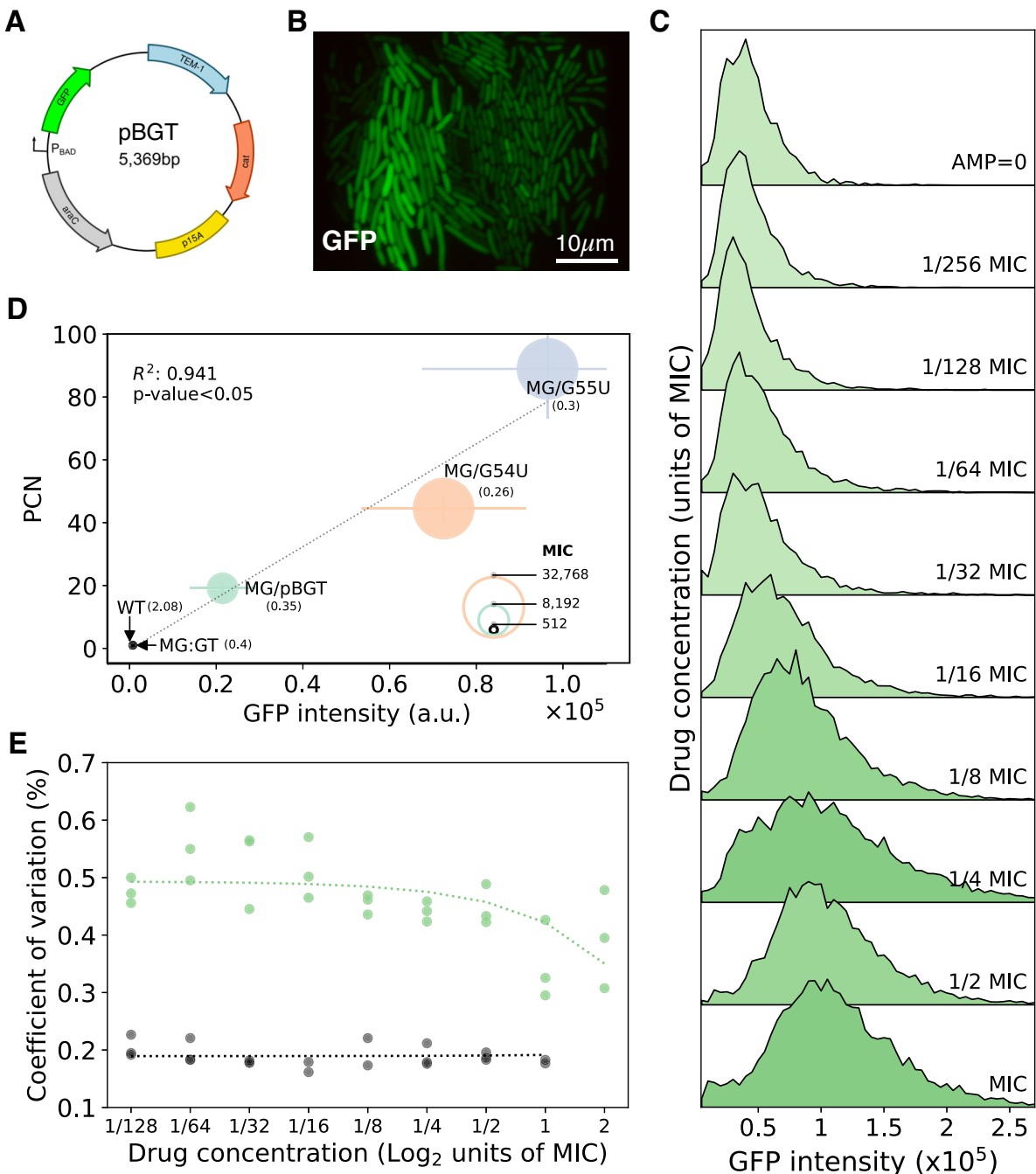

**Fig. 2 | Experimental model system. A** Schematic representation of plasmid pBGT encoding *bla*TEM-1 (in blue) and *GFPmut2* (in green). The reading frames for genes are represented with arrows, with arrowheads indicating the direction of transcription. **B** Representative fluorescence microscopy image of the plasmid-bearing population (MG/pBGT) shows high levels of GFP heterogeneity between cells. The GFP distribution was obtained after analyzing 46 separate microfluidic chambers. **C** Fluorescence distributions of MG/pBGT exposed to a range of AMP concentrations. **D** Mean fluorescence and mean plasmid copy number are positively correlated in bacterial populations in the absence of selection. The circle's diameter is proportional to each strain's drug resistance level. Black dotted line is a linear regression of PCN vs GFP intensity ($R^2 = 0.941$, $p$ value < 0.05). The numbers in parentheses next to each name represent the Coefficient of Variation (CoV) in GFP intensity. **E** Coefficient of variation of GFP distributions in response to a range of drug concentrations. Data for MG/pBGT is denoted with green circles and for MG:GT in black circles. Dotted lines represent the linear regression performed over the range of drug concentrations.

compensatory mutations, another strategy to ameliorate the burden of carrying high-copy plasmids is to reduce the number of plasmids carried per cell. For instance, a previous experimental evolution study reported that mutations near the origin of replication generated a 10-fold amplification in mean PCN, but at a very high fitness cost that resulted in high levels of antibiotic resistance being unstable in the population once the antibiotic was removed from the environment[27].

To experimentally determine how rapidly PCN amplification reverses once the antibiotic is no longer present, we conducted a three-season serial dilution experiment in which a MG/pBGT population was subjected to fluctuating selection (season 1, no drug; season 2, 32 mg/ml AMP; season 3, no drug). The distribution of GFP fluorescence was recorded at the end of each season (Fig. 3A). As anticipated by the model, the distribution of GFP fluorescence shifted to high

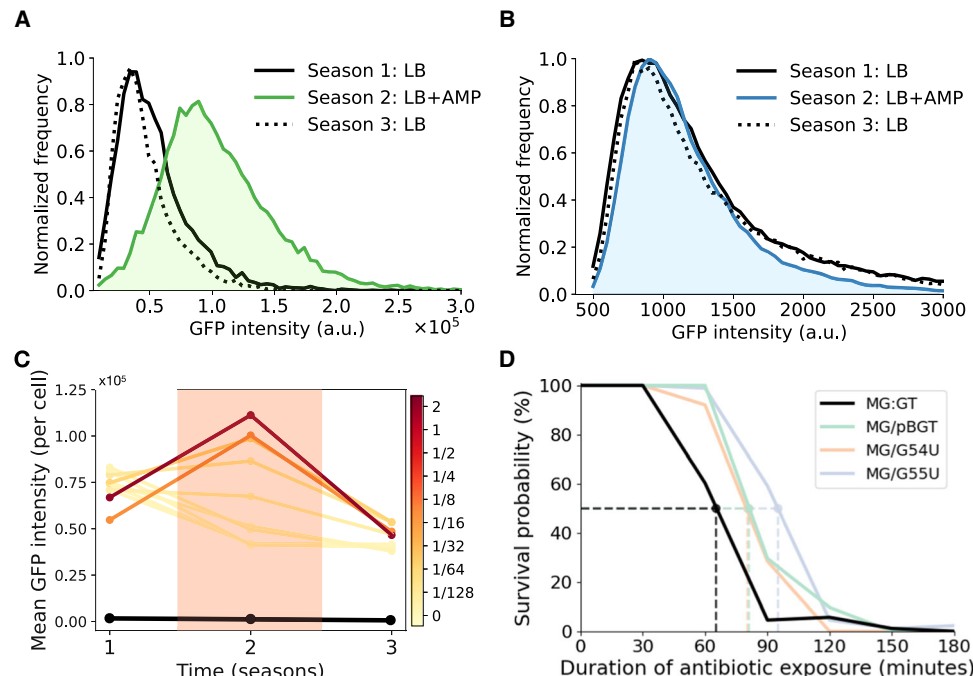

**Fig. 3 | Effect of antibiotics on GFP and survival under fluctuating selection.**
**A** GFP histogram in a population of MG/pBGT exposed to 12-h season treatments (Season 1 (LB): solid black line, season 2 (LB+AMP): green area/line, season 3 (LB): dotted black line). Note that the antibiotic shifts the GFP distribution to the right (green area) and is later restored when the antibiotic is removed. **B** GFP histogram for MG:GT reveals that GFP distributions coincide independently of the environmental drug concentration. **C** Increase in mean fluorescence in the presence of antibiotics is correlated with drug dose (darker red, higher drug concentrations).

Once the antibiotic is removed, mean GFP intensity is restored to pre-exposure levels. The black line shows that fluorescent intensity for MG:GT remains constant during the experiment. **D** Kaplan−Meier plot comparing survival probabilities as a function of the time exposed to a lethal ampicillin concentration (with MIC determined separately for each strain). Dotted lines represent the duration of drug exposure that results in a 50% survival probability (MG:GT in black, MG/pBGT in green).

expression levels in the presence of AMP, but quickly returned to the original distribution once the antibiotic was removed. This effect was also observed with high-copy plasmids (Fig. S2), but not in the presence of the $\beta$-lactamase inhibitor sulbactam (Fig. S4).

As predicted by our computer simulations, repeat runs of the experiment with different AMP concentrations showed that the mean GFP fluorescence of the MG/pBGT population increased proportionally to the strength of selection, and the shift towards higher copy numbers was quickly reversed after the antibiotic was removed. In contrast, the GFP intensity distribution in MG:GT cultures was unchanged, regardless of the presence of antibiotic in the medium (Fig. 3B). Altogether, we conclude that multicopy plasmids offer a versatile platform for rapid and elastic amplification and attenuation of gene dosage, both experimentally (Fig. 3C) and computationally (Fig. 1G).

## PCN variability enhances survival of populations exposed to fluctuating selection

We examined the response to AMP of 88 clonal populations of plasmid-bearing strains with different mean PCNs: MG/pBGT, MG/G54U and MG/G55U, with 19, 44, and 88 plasmid copies, respectively. Once the cultures reached exponential growth, we transferred approximately 1% of each population to a replenished media environment containing a lethal concentration of AMP. After 30 min, a sample of each population was returned to drug-free medium, and this sampling process was repeated every 30 min. For each duration of drug exposure, we counted the number of replicates exhibiting growth after 24 h.

Our main finding is that plasmid-bearing strains exhibited a significant survival advantage relative to non-plasmid-bearing strains in the face of fluctuating selection (Fig. 3D; log-rank test, $p$ value < 0.005). For example, after 90 min of AMP exposure, the probability of survival was > 50% for all plasmid-bearing strains, whereas < 5% of the non-

plasmid-bearing populations survived. It should be noted that the lethal drug concentration was determined independently for each strain (see Table S1 for MICs used).

To rule out the possibility that the increased resistance exhibited by plasmid-bearing strains was due to a decrease in growth rate associated with the metabolic burden of carrying plasmids, rather than selection of a subpopulation with more copies of $bla_{TEM-1}$, we performed a survival assay for MG/pBGT in the presence of 256 µg/L of the $\beta$-lactamase inhibitor, sulbactam. As expected, fluorescence remained constant in environments where sulbactam removed the selective advantage of carrying the plasmid, confirming our hypothesis that the survival benefit is due to selection of subpopulations with higher $bla_{TEM-1}$ copy numbers rather than a consequence of the metabolic burden of carrying plasmids (Fig. S5A).

In the absence of sulbactam, we observed a significant increase in mean PCN in all plasmid-bearing populations, consistent with our previous findings. This suggests that a subpopulation of highly-resistant cells with increased PCN is responsible for the observed survival advantage during an antibiotic pulse. Moreover, we found that PCN was positively correlated with both the rate of increase in per-cell GFP abundance (Fig. S5A) and the median survival time of each strain, defined as the duration of drug exposure required to achieve a 50% probability of survival (Fig. S5B). For MG:GT at 2 mg/mL AMP, the median survival time was 60 min, while for MG/pBGT at 32 mg/mL AMP, it was 80 min. We refer to this duration of exposure and concentration as a *semi-lethal pulse*.

## Single-cell measurement of fluorescence and survival to a semi-lethal pulse

In a microfluidic chemostat, MG/pBGT and MG:GT populations were exposed separately to a semi-lethal pulse of AMP (Fig. 4A).

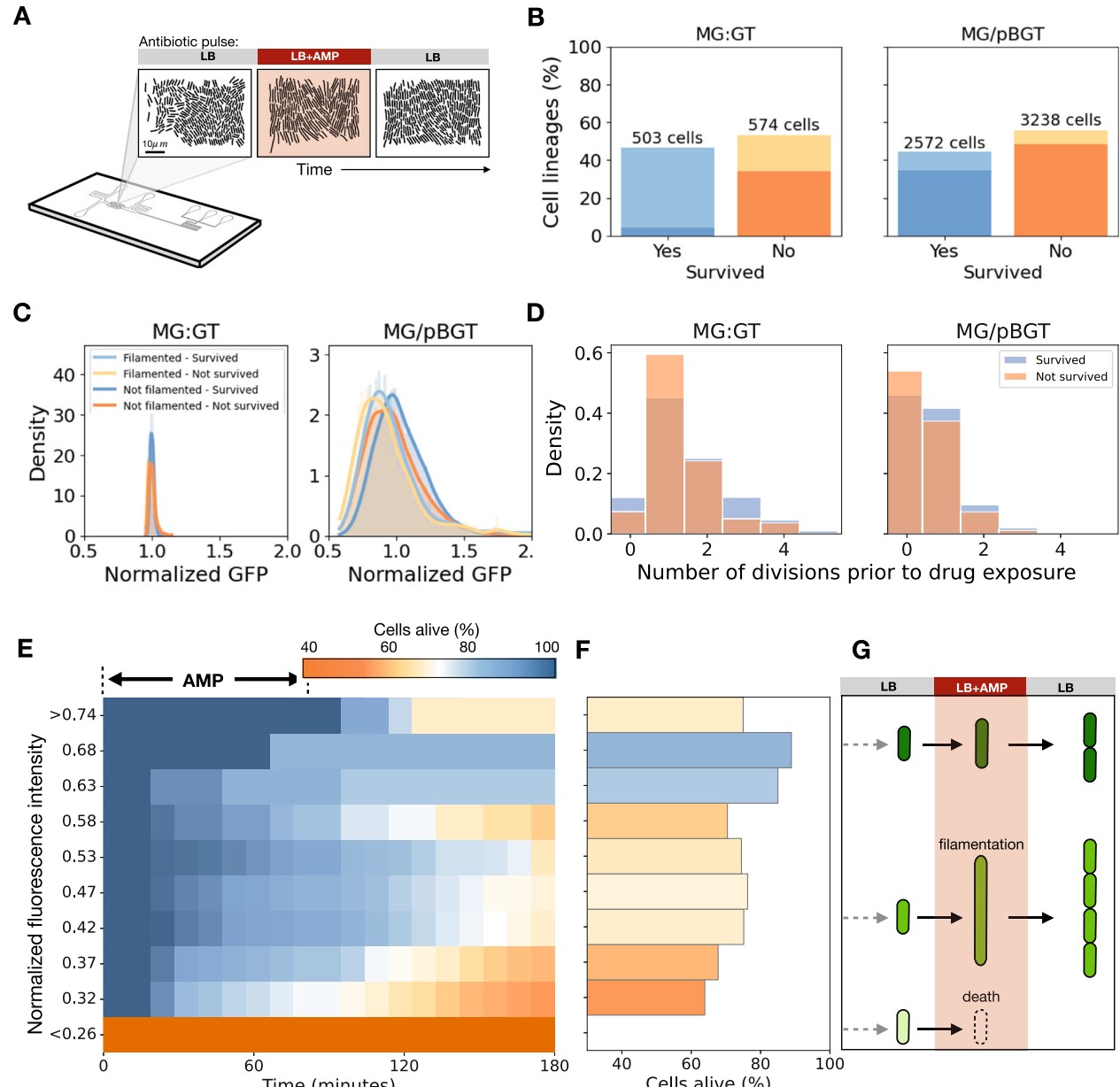

**Fig. 4 | Single-cell analysis of a semi-lethal pulse. A** Schematic diagram illustrating a microfluidic experiment exposing MG:GT and MG/pBGT populations to a semi-lethal antibiotic pulse. **B** Results obtained after tracking individual cell lineages in time-lapse movies. Cell lineages are classified based on whether they survived treatment (orange and blue) or if they died during drug exposure. **C** GFP distributions of cells in MG:GT (left) and MG/pBGT (right) show significant differences in mean GFP and variance. For MG:GT, the fluorescent distribution is characterized by low variance and no significant differences in mean GFP between cells that produced filaments and were killed (light orange) or survived (light blue), as well as for cells that did not produce filaments and died (dark orange), and those that survived drug exposure (dark blue). The plasmid-bearing population exhibits a large variance in GFP distribution, with survived cells showing an increased mean fluorescence relative to cells killed. For surviving cells, mean GFP was significantly lower for cells that did not produce filaments than cells that filamented.

**D** Histogram shows the number of division events per cell lineage in the MG/pBGT (right) and MG:GT (left) populations prior to drug exposure. **E** Fraction of cells alive as a function of time for lineages present upon antibiotic introduction. The y-axis denotes the initial fluorescence of cells in each initial GFP bin, and each box represents the proportion of cells that are still alive in each time step (high survival rates in a light color). **F** Histogram of GFP expression for MG/pBGT cells estimated after drug exposure. The size of each bar represents the probability of survival estimated for each GFP level after exposure to a semi-lethal pulse of AMP. Note how the distribution appears bimodal, with high survival rates at intermediate and very high fluorescent intensities. **G** Diagram illustrating that this bimodal distribution results from a stress response mechanism producing filamented cells and provides transient resistance to ampicillin in cells with intermediate fluorescent values. Cells with low GFP values before drug exposure have a low probability of survival, while cells with high fluorescent intensities are highly resistant to the antibiotic.

We recorded the fluorescent intensity and division events of individual cells in time-series data and estimated the cell-division rates of 5810 lineages for MG/pBGT and 1077 lineages for MG:GT. These data were obtained from 46 and 8 separate microfluidic chambers, respectively (see Supplementary Movies 1 and 2 for sample time-lapse movies).

For our analysis, we included single-cell lineages that were observed during the entire duration of the antibiotic pulse.

First, our microfluidics data show that, for both MG/pBGT and MG:GT, the fluorescence of surviving cells remained constant throughout the antibiotic exposure period (Fig. S6). This indicates that

the increase in mean GFP observed in the MG/pBGT population is due to the antibiotic selectively eliminating low-copy-number cells, rather than individual cells upregulating plasmid replication or $bla_{TEM-1}$ expression. To determine the extent of AMP-induced cell lysis, we stained the medium with rhodamine and measured the intracellular accumulation of the fluorescent dye. This enabled us to differentiate cells that survived from those that were killed in response to the antibiotic (Figs. S7 and S8). We expected the semi-lethal pulse to kill approximately half of the population, as the antibiotic concentration and duration of treatment were determined separately for each strain. In agreement with this prediction, we found that 46.7% of MG:GT cells and 44.2% of MG/pBGT cells survived the antibiotic pulse (Fig. 4B).

A retrospective analysis of surviving and non-surviving cells revealed that surviving cells had an elevated duplication rate (Fig. 4D; $p < 0.005$), (in average 38.7 and 87.1 min between cell duplication events, respectively). Similarly, surviving cells had a higher rate of elongation (changes in cell length between consecutive frames) than cells that were killed (Fig. S9; $p < 0.05$). These results suggest that an enhanced probability of survival is a consequence of a dosage effect and heterogeneity in $bla_{TEM-1}$ expression and not of reduced metabolic activity associated with bearing plasmids.

The relationship between GFP expression before drug exposure and survival is shown in Fig. 4E. Cells with high GFP expression had a larger probability of survival (54% survival for the top quartile), while cells in the bottom quartile had a mean survival rate below 34%. Interestingly, survival probability was not a monotonously increasing function of GFP intensity, as high survival rates were also observed in cells with intermediate GFP expression (Fig. 4F). In the following section, we will show this is a consequence of cells with intermediate GFP abundances that survived exposure to the antibiotic by elongating and delaying their cell division. For MG:GT, we found that the probability of survival remained constant, independently of the GFP intensity exhibited by each cell prior to AMP exposure (Fig. S10).

### Plasmid-driven phenotypic noise produces a heterogeneous stress response

Exposure to the antibiotic resulted in some cells that stopped dividing but continued to grow, leading to the production of filaments[53,54] (Fig. 4G). We defined a filamented cell as one with a length greater than two standard deviations from the mean length of the population before drug exposure. We classified each cell according to whether it survived drug exposure, and whether or not it produced filaments, and observed that most surviving cells in the MG:GT population produced filaments (Fig. 5A. B). Drug-induced filamentation can be triggered by multiple molecular mechanisms[55], including a two-component signal transduction system[56] that blocks FtsZ polymerization and prevents septation in the presence of AMP. After the stress is removed, filamented cells rearrange the FtsZ ring, divide, and resume normal growth[57,58].

Our data support earlier results showing that stress response machinery is tightly regulated[59]. When exposed to the antibiotic, 61.4% of MG:GT cells synchronously produced filaments (Fig. 5C, D), in contrast to the plasmid-bearing population that exhibited a highly heterogeneous response, with only 17.1% of cells producing filaments (Fig. 5E, F). The heterogeneity in the response of the MG/pBGT population observed experimentally (Fig. 5G, H) was anticipated, as the computational model predicted that PCN variation resulted in the existence of cells that overproduce $\beta$-lactamase, thereby maintaining a low periplasmic concentration of AMP and delaying the triggering of the stress response. The model also predicted that cells with low PCN would be killed by the antibiotic before they had a chance to respond, and the experimental data confirmed this behavior.

For each subpopulation, Fig. 4C shows the histograms of GFP fluorescence prior to the introduction of AMP into the microfluidic device. Consistent with the population-level experiments, the MG:GT population exhibited low variance, in contrast to the plasmid-bearing population that presented a GFP intensity distribution with large variance. We classified each cell according to whether it was killed or survived drug exposure and according to whether or not filaments were produced, and found that most surviving cells in the MG:GT population produced filaments (Fig. 4B). Furthermore, we found no correlation between GFP fluorescence and survival in this population (Fig. S10).

On the other hand, among the MG/pBGT population, the surviving cells displayed two distinct characteristics: either they had a high fluorescence intensity and exhibited slow growth, or they had intermediate GFP fluorescence and formed filaments (Fig. 4C, F; Fig. S11). An exploratory data analysis revealed that both PCN (measured indirectly through GFP intensity) and cell length at the time of the environmental perturbation were crucial for cell survival (see the PCA plot in Fig. S12). Overall, these results suggest that plasmid-driven phenotypic noise promotes random conditional filamentation, enabling the population to survive to fluctuating selection by implementing a bet-hedging strategy[60,61].

## Discussion

The evolution of antimicrobial resistance in response to the industrialized consumption of antibiotics, specifically those of the $\beta$-lactam class, is one of the most serious health threats societies face today[62]. While drug resistance has traditionally been attributed to stable genetic mutations or horizontal gene transfer of resistance genes, resistance can also arise through genomic duplications that increase the dosage of known drug-resistance genes[7,63,64], such as amplification of efflux pump operons[65] or genes encoding drug-modifying enzymes[66,67].

Laboratory studies have demonstrated that genomic amplifications increase in response to stronger selective pressure[63], but are unstable without continued selection due to the fitness burden associated with duplicating large chromosome regions[63,68,69]. In the clinic, tandem duplications of known antibiotic resistance genes have been shown to produce high levels of heteroresistance in clinical isolates from multiple bacterial species evaluated against a range of antibiotics[67,70,71], even in strains that were considered antibiotic-sensitive by conventional clinical methods[72]. Notably, while some of these duplications occurred in large chromosomal regions containing known drug-resistance genes, others were found in plasmids[5].

Here we investigated the role of multicopy plasmids in the amplification of the antibiotic resistance gene $bla_{TEM-1}$ and the emergence of transient antibiotic resistance. Our study used stochastic simulations of a computational model and high-throughput single-cell measurements of gene expression in E. coli MG1655 to show that PCN variability maintains a stable and diverse population with a wide range of plasmid copy numbers. Furthermore, we found that the distribution of plasmid copy numbers was responsive to the environment, with the mean copy number increasing rapidly in the presence of selection, and with a degree of amplification that was proportional to the strength of selection.

The rapid increase in PCN observed in populations under selective conditions can be explained by two possibilities: a uniform increase in resistance levels across all cells in the population (e.g., by enhancing the rate of plasmid replication) or cell-to-cell heterogeneity in resistance levels (pre-existing PCN variability in the population). Population-level experiments cannot distinguish between these hypotheses, so we used single-cell microfluidics to monitor GFP abundance of individual cells and their corresponding phenotypic state in response to AMP exposure. Using quantitative image analysis and computer simulations, we confirmed that selection of highly-resistant cells drives PCN amplification, indicating that pre-existing PCN variability is responsible for the rapid increase in resistance levels observed in the population in the presence of drug.

Similarly, our computer experiments show that different PCN distribution shapes, influenced by diverse noise sources including replication, segregation, and copy number control, lead to

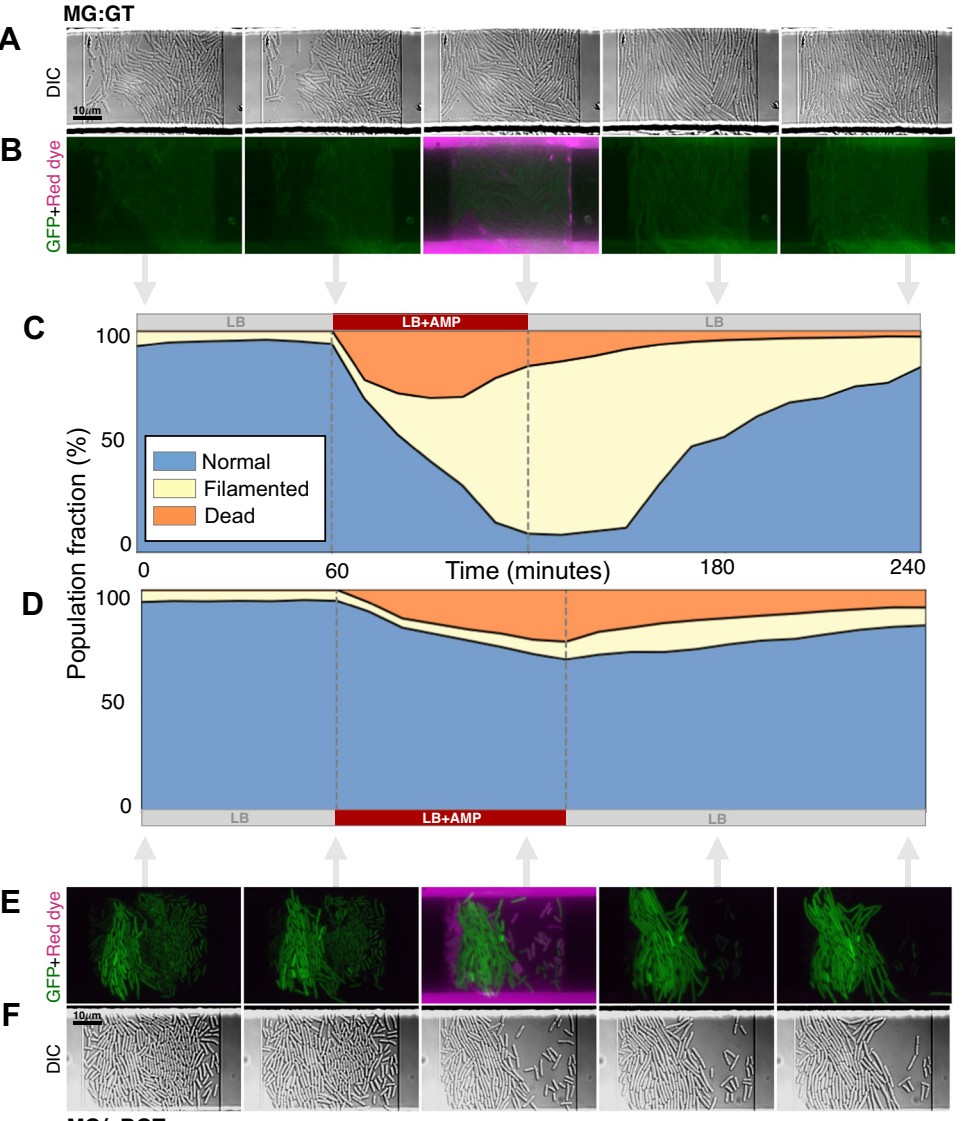

**Fig. 5 | Microscopy montage of a microfluidics semi-lethal pulse. A** Montage of a representative time-lapse images of MG:GT growing in a microfluidic device exposed to a semi-lethal pulse of AMP. We obtained data from 5810 lineages from 46 microfluidic chambers. **B** Overlay time-lapse movie showing the fluorescent intensities of GFP (green) and rhodamine (magenta) in the MG:GT population. **C** Fraction of the MG:GT population in each cellular state as a function of time (normal cells in blue, stressed in yellow, and dead in orange). Most surviving cells exhibit conditional filamentation upon antibiotic exposure and resume normal growth once the drug is withdrawn. **D** Fraction of the MG/pBGT population in each cell state. In this case, a smaller fraction of cells produce filaments, as high PCN cells maintain low periplasmic levels of antibiotics and survive without triggering the stress response system. **E** Overlay of selected frames from the time-lapse movie shows variable GFP expression levels within the population, as well as varying response to the drug exposure. **F** DIC images of selected frames of the MG/pBGT population growing in a microchemostat. We obtained data from 1077 lineages from 8 separate microfluidic chambers.

heterogeneous populations (Supplementary Information). Importantly, this population heterogeneity was associated with enhanced resilience under fluctuating selection, including improved survival against rapid antibiotic introduction and better adaptation to antibiotic withdrawal.

Our study focused on non-conjugative, multicopy plasmids that are usually carried at around 10–30 copies per cell; however, plasmid-driven phenotypic noise is not exclusive to high-copy plasmids[73]. A recent study showed that conjugative, low PCN populations (1–8 copies per cell) also exhibited large copy number heterogeneity that resulted in noisy expression of plasmid-encoded genes[74]. Furthermore, transient amplification of selective genes encoded in multicopy plasmids may not be exclusive to $bla_{TEM-1}$, as similar effects would be achieved by antimicrobial resistance genes encoding efflux proteins or other drug-modifying enzymes[7,75–77]. In conclusion, our results provide insights into

how bacteria can transiently survive lethal concentrations of antibiotics by using multicopy plasmids as a platform for rapid amplification and attenuation of gene copy numbers. Understanding the role of multicopy plasmids in antibiotic resistance is critical, as these elements not only act as vehicles for the horizontal transfer of genetic information between cells but also facilitate bacterial adaptation in dynamic environments.

## Methods
### Bacterial strains and culture conditions
In this study, we used *Escherichia coli* K12 MG1655 bearing a ColE1-like (p15A) plasmid, pBGT, encoding for the $\beta$-lactamase resistance gene $bla_{TEM-1}$ that confers resistance to ampicillin under a constitutive promoter, an *GFPmut2* gene under an arabinose inducible promoter, and the *araC* repressor. Mean PCN = 19.12, s.d. = 1.53[36]. As a control, a strain E. coli K12 MG1655 was used, carrying the construct

$araC - pBAD - gfp2 - bla_{TEM-1}$ inserted into the chromosome at the $\lambda$-phage integration site ($attB$). Strains bearing plasmid variants G54U and G55U contained a point mutation in the origin of replication: G to U changes at positions 54 and 55 of the RNAI placed in the loop of the central hairpin and affect the RNAI-RNAII kissing complex that controls plasmid replication and PCN. All experiments were conducted in Lysogeny Broth- Lenox (LB) (Sigma-L3022) supplemented with arabinose (0.5% w/v) and appropriate ampicillin concentrations were supplemented as indicated in each experiment. Arabinose stocks solutions were prepared at 20% w/v by diluting 2 g of arabinose (Sigma-A91906) in 10 ml DD water sterilized by 0.22 μm filtration. AMP stock solutions (100 mg/ml) were prepared by diluting ampicillin (Sigma-A0166) directly in 0.5% w/v arabinose LB.

## Antibiotic susceptibility determination

The minimum inhibitory concentration (MIC) of different strains was calculated using dose-response curves performed in 200 μL of liquid media. 96-well plates (Corning CLS3370) supplemented with LB (0.5% w/v arabinose) and a logarithmically-separated range of drug concentrations were used. Antibiotic plates were inoculated from a master plate using a 96-pin microplate replicator (Boekel 140500). Inoculation plates were prepared by adding 200 μL of overnight culture into each well and incubating at 37 °C with 200 rpm shaking. Optical density measurements were performed using a BioTek ELx808 Absorbance Microplate Reader at 630 nm. MIC was determined when the reader was unable to detect bacterial growth (2, 32, 43, and 46 mg/mL for strains MG:GT, pBGT, G54U, and G55U, respectively).

## Plasmid copy number determination

PCN per chromosome was determined using quantitative polymerase chain reaction (qPCR) with a CFX96 Touch Real-Time PCR Detection System. Specific primers were used for the E. coli's $dxs$ monocopy gene as chromosomal reference (dxs-F CGAGAAACTGGCGATCCTTA, dxs-R CTTCATCAAGCGGTTTCACA) and primers for the $bla_{TEM-1}$ plasmid-encoded gene (Tem-F: ACATTTCCGTGTCGCCCTT, Tem-R: CACTC GTGCACCCAACTGA) both with amplicon sizes 100 bp as previously described[36]. In short, samples were prepared following a previously published protocol[78]: 100 μl culture samples were centrifuged at 16,000 g for 60″, the supernatant was removed, and the pellet was resuspended in an equal volume of MilliQ water. Then, samples were boiled at 95 °C for 10′ using a thermoblock and stored at −20 °C for later use. Primers were diluted in TE buffer at 10 μM and stored a −20 °C. Primers' final concentration was 300 nM. qPCR reactions were performed using SYBR Select Master Mix (Applied Biosystems - 4472908) in 96-well flat-bottom polystyrene microplates (Corning 3370) sealed with sterile optical film (Sigma-Aldrich Z369667-100EA). Amplification was performed by an initial 2 min at 50 °C activation, then an initial denaturation for 2 min at 95 °C, following 40 cycles of 15 s denaturation at 95 °C, 1 min annealing, and 1 min extension at 60 °C. After the amplification, a melting curve analysis was performed by cooling the reaction to 60 °C and then heating slowly to 95 °C. PCN was determined using the $\Delta\Delta C_T$ method[79].

## Flow cytometry

GFP fluorescence distributions were calculated using imaging flow cytometry in an Amnis ImageStream Mark II by Luminex. INSPIRE software was used to control the machine and acquire data. GFP fluorescence was excited at 488 nm using 75 mv intensity. Data files were processed using IDEAS 6.2 software to only take into account cells on focus using area, aspect ratio, and side scatter features. Files were exported to text files and analyzed with custom scripts in Python. Fluorescence-activated cell sorting of the MG/pBGT strain using a BD FACSAria. An overnight culture was grown on 20 ml of LB 0.5% w/v arabinose at 30 °C, and 200 rpm was sorted into subpopulations. Four subpopulations were categorized by their fluorescence intensity and SSC-area features. DNA extraction of sorted subpopulations was made as previously described for qPCR and stored at −20 °C for later use. Plasmid copy number measurements were performed in each subpopulation to evaluate the association between copy number and fluorescence intensity.

## Fitness costs determination

To determine strains fitness in the absence of antibiotics, each strain was cultured in a 96-well plate with LB supplemented with arabinose 0.5% w/v. A Synergy H1 microplate reader was used to obtain the growth kinetics of each strain by inoculating a 96-well plate with an overnight culture of each strain and growing at 37 °C for 24 h, reading every 20 min, after 30 s of shaking. Maximum growth rate estimates were obtained by fitting the mean optical density of $N = 8$ using the R package $GrowthRates$ using non-parametric smoothing splines fit[80].

## Semi-lethal pulse in bacterial populations

Strains of MG:GT and MG/pBGT were exposed to a three-season serial transfer experiment using 96-well plates (8 replicates per strain). An initial inoculation plate was made by putting 200 ml of overnight culture per well. Season 1 (LB) was inoculated from an inoculation plate using a microplate pin replicator. Season 2 (LB-AMP) was inoculated from season 1 after 12 h of growth. We used the following ampicillin gradient: 0, 1/128, 1/64, 1/32, 1/16, 1/8, 1/4, 1/2, 1, and 2 MIC units. In season 3, cultures were transferred to a new LB plate after 12 hours of growth, allowing bacteria to grow for another 12 h. Plates were sealed using an X-Pierce film (Sigma Z722529) perforating every well to avoid condensation and grown at 37 °C inside a BioTek ELx808 Absorbance Microplate Reader. Measurements were taken every 20 min, after 30 s of linear shaking at 567 cpm (3 mm). At the end of each season, end-point fluorescence intensity was measured using a BioTek Synergy H1 using OD (630 nm) and eGFP (479.520 nm). Plates were then stored at 4 °C before imaging flow cytometry was performed the following day. A complete independent four-replicate experiment was performed for each strain. DNA samples were extracted at the end of each season to quantify PCN.

## Population-level survival assay

Strains were grown in an LB+Amp media in a 96-well plate under a concentration of AMP determined based on the MIC of each strain. For each LB+AMP plate, we considered 88 populations growing in antibiotics and 8 without antibiotics as controls. Inoculated plates were incubated in a BioTek ELx808 absorbance microplate reader at 30 °C, with optical density measurements (630 nm) obtained every 30 min, after 1 min of shaking. After each read, plates were taken out, and a plate sample was taken with a microplate replicator to inoculate a new LB plate. Samples were taken every 30 min, from 0 to 8 h, then at 18 and 24 h. New plates were grown in a static incubator at 30 °C for 24 h. Growth was measured using OD (630 nm) and eGFP (479,520 nm) in a Synergy H1 microplate reader after 5 min shaking. An additional experiment was performed for the MG:GT and MG/pBGT strains sampling every 2 h from 0 to 12 h and a final sampling at 24 h.

## β-lactamase inhibitor experiment

For the $\beta$-lactamase inhibition assay, sulbactam (Sigma-S9701) was used. First, the ampicillin concentration was fixed to be that of the MIC of MG:GT (2 mg/ml). Then, a sulbactam dose-response experiment with MG/pBGT was performed and found that the minimum sulbactam concentration achieved that complete growth suppression was 256 μg/l. Critical AMP and sulbactam concentrations were used to perform a population-level survival assay consisting on exposing 8 replicate populations to fluctuating selection: LB → LB +AMP+sulbactam → LB. Samples of four replicates were used for flow cytometry, and the remaining four replicate samples were used for PCN quantification.

## Single-cell microfluidics

A microfluidic device built-in PDMS (polydimethylsiloxane; Sylgard 04019862) from molds manufactured by Micro resist technology GmbH using soft photolithography (SU-8 2000.5) was used for this study. In particular, a micro-chemostat that contains two media inputs and 48 rectangular chambers ($40 \times 50 \times 0.95\ \mu m^3$)[81]. Each confinement chamber traps approximately 1000 cells in the same focal plane, enabling us to use time-lapse microscopy to follow thousands of individual cells in time. Chips were fabricated by pouring PDMS into the mold before baking it for 2 h at 65 °C. Solid chip prints were cut, punched, and bound to a glass coverslip using a plasma cleaner machine (Harrick Plasma - PDC-001) at full power for 1 min and 15 s. Then we baked them again overnight at 45 °C to ensure binding. Moreover, for each strain, MG/pBGT and MG:GT, a 1$l$ titration flask was inoculated with 200 µl of an overnight culture (LB at 30 °C and 200 rpm) when the culture reached 0.2–0.3 OD630; it was split into 4 falcon tubes and centrifuged for 5 min at 7 rpm. The supernatant was disposed of, and cells were resuspended by serial transfers into 5 ml of fresh media supplemented with arabinose 0.5% w/v. This dense culture was used to inoculate the microfluidic device. Data acquisitions started 5 h after the device chambers were filled and cells were growing exponentially.

After 60 min of growth, we switched the environment from LB to LB+AMP. Drug concentration was determined independently for each strain (2 mg/ml and 8 mg/ml for MG:GT and MG/pBGT, respectively). Media and antibiotics were introduced into the microfluidic device using a bespoke dynamic pressure control system based on vertical linear actuators (adapted from[82]). The duration of drug exposure was determined based on the time elapsed before the probability of survival of the population exposed to the MIC is below 50% (a semi-lethal pulse; an exposure of 2 mg/ml for 60 min for MG:GT, and of 8 mg/ml for 80 min for MG/pBGT). At the end of the period of drug exposure, the population was transferred to a drug-free environment and grown for 120 min for MG:GT, and 100 min for MG/pBGT. Growth media was supplemented with arabinose at 0.5% and Tween20 (Sigma-P2287) at 0.075%, and filtered with 0.22 µm filters. Experiments were conducted at 30 °C, and the ampicillin media was stained by adding 5 µl and 3 µl of a fluorescent dye (rhodamine, Sigma S1402) in 100 ml of media used to grow MG:GT and MG/pBGT cells, respectively. This red fluorescent dye allowed us to calibrate media inputs inside the microfluidic device and also worked as a dead-cell marker. Rhodamine stock solution was prepared, diluting the powder in ethanol, and stored at 4 °C.

## Fluorescence microscopy

Microscopy was performed in a Nikon Ti-E inverted microscope equipped with Nikon's Perfect Focus System and a motorized stage. Temperature control is achieved with a Lexan Enclosure Unit with Oko-touch. The microscope was controlled with NIS-Elements 4.20 AR software. Image acquisition was taken with a 100x Plan APO objective without analog gain and with the field and aperture diaphragms as closed as possible to avoid photobleaching. DIC channel captures were made with a 9v DIA-lamp intensity, red channel (excitation from 540 to 580 nm, emission from 600 to 660 nm filter), green channel (excitation from 455 to 485 nm, emission from 500 to 545 nm). Exposure times were 200 ms, 200 ms, and 600 ms for DIC, green and red channels, respectively. Multi-channel, multi-position images were obtained every 10 min in the following order: Red, Green, Lamp-ON, DIC, Lamp-OFF. We added the Lamp-ON optical configuration to allow the bright-light lamp to be fully powered before acquiring the DIC image, while the Lamp-OFF configuration was added to make sure that the lamp was completely off before capturing the next position.

## Image analysis

Microscopy time-lapse images were analyzed using µJ, an ImageJ-Python-Napari image analysis pipeline that implements Deep Learning for image segmentation. In short, the pipeline uses ImageJ macros to arrange and manipulate microscopy images. Image segmentation was performed using DeepCell[83]. Binary masks were corrected manually using bespoke ImageJ macros. Cell tracking was performed using a nearest-neighbor weighted algorithm coded in Python. Cell-tracking was corrected manually using a custom cell viewer coded in Napari. Lineage reconstruction was performed in Python, obtaining thousands of single-cell time-series of fluorescent intensity and cell length, as well as time-resolved population-level statistics, including the probability of survival to the antibiotic shock and the distribution of fluorescent intensities.

## Computational model

We developed a stochastic individual-based model where cells are modeled as computational objects. The complete and comprehensive description of the model is provided in the Supplementary Information. The numerical experiments based on this model were implemented using Julia, and the corresponding code has been made publicly available in a repository: https://github.com/ccg-esb-lab/pBGT/.

## Reporting summary

Further information on research design is available in the Nature Portfolio Reporting Summary linked to this article.

## Data availability

The data used in our study is available in the project's public repository: https://doi.org/10.5281/zenodo.10403406.

## Code availability

The code used to analyze images is available in a public repository: https://github.com/ccg-esb-lab/uJ/ Code necessary for simulating the computational model can be found in the project's public GitHub repository: https://doi.org/10.5281/zenodo.10403406.

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

## Acknowledgements

We thank S. Brom, J. Stavans, D. Romero, P. Padilla, M. Ackermann and R. Beardmore for useful discussions and comments on earlier versions of this manuscript. We are also thankful to A. Saralegui from Laboratorio Nacional de Microscopía Avanzada for assistance using the flow cytometer and J. Escudero for the gift of the strains. We also thank the LABNALCIT-UNAM (CONACYT) for technical support using the cell sorter. JCRHB was a doctoral student in Programa de Doctorado en Ciencias Biomédicas, Universidad Nacional Autónoma de México, and received fellowship 59691 from CONACYT. BAL is a student in Programa de Doctorado en Ciencias Bioquímicas, Universidad Nacional Autónoma de México and received fellowship 886346 from CONACYT. J.V.S. received a scholarship from PAPIIT-UNAM (grant IN209419). A.S.M. is funded by the European Union's Horizon 2020 research and innovation program (ERC grant agreement no.757440-PLASREVOLUTION). J.R.-B. is supported by a Miguel Servet contract from Instituto de Salud Carlos III (ISCIII; grant no. CP20/00154), co-funded by ESB, 'Investing in your future'. R.P.M. and R.C.M. were supported by a Newton Advanced Fellowship awarded by the Royal Society (NA140196). A.F.H. and R.P.M. were supported by PAPIIT-UNAM (grants IA201418 and IN209419, respectively). This project was also funded by CONACYT Ciencia Básica (grant A1-S-32164) awarded to R.P.M.

## Author contributions

R.P.M., A.F.H., J.R.B., A.S.M and R.C.M. conceived and planned the experiments. J.C.R.H., B.A.L. and A.F.H. carried out the experiments. J.C.R.H., J.V.S., B.A.L. and R.P.M. analyzed the data. O.M.P., J.C.R.H., R.P.M. and A.F.H. developed the microfluidics platform. J.C.R.H. and R.P.M. implemented and postulated the computational model. J.C.R.H., A.F.H. and R.P.M. wrote the manuscript. All authors discussed the results and commented on the manuscript.

## Funding

## Competing interests
