## [Peer Review File · Nature Communications]

Plasmid-mediated phenotypic noise leads to transient antibiotic resistance in bacteriaREVIEWER COMMENTS

Reviewer #1 (Remarks to the Author):

This is a very high-quality and detailed paper, that addresses the consequences of plasmid copy number variability in multicopy plasmids on population-level heterogeneity and response to antibiotic selection. The authors have previously studied in detail other aspects of copy number dynamics; here they focus on fluctuating environments. The results are nicely discussed in the context of both plasmid biology and phenotypic heterogeneity.

I have a few comments below, overall minor, in order of appearance in the text:

- L63 'an adaptive strategy' – for whom? Which level of selection is this?

- L67 encoding genes is beneficial

- L71 and elsewhere: maybe avoid the use of the word 'tolerance' in this manuscript, as this is used to describe mechanisms that do not involve changes in antibiotic resistance itself, whereas the phenomenon described here fully relies on resistance gene expression.

- Figure 1: "A)" is lacking from the legend text; the gradient of greens in 1A could be added as a color legend to make more obvious which side of the gradient corresponds to high PCN. In 1F, I can see a red line, but no black line and no dotted line described in the legend.

- Figure 3: could the authors explain, or comment on why there is no difference in survival between pBGT and G54U populations despite the higher copy number of G54U?

- Movies S1, S2 and Fig 5C: why does one of these populations have no red (dead) cells at all, if the semi-lethal antibiotic concentration has been normalised so that each strain experiences the same amount of death?

- Figure 4D shows the number of divisions whereas the text L232 measures the time elapsed between cell division events, using the same metric for both would be easier to follow for the reader. Similarly, Figures 4E and F are divided into 9 bins of GFP intensity but the text mentions quartiles instead.

- L304: enabling the population to survive

- L324 and whole paragraph: Fig 5C does not really make the point about cell division time described in the text. I wonder if it wouldn't also be worth introducing plasmid cost (Fig S13) earlier, even within Fig 2 together with Fig 2D, as part of the initial characterisation of the system.

- L346: I did not understand the point the authors are making about decay of selection, or how this is visible in Fig 6D.

- The whole section page 13 of the manuscript is a bit difficult to follow, it feels like subsections of figures have been moved around too many times, and it would be easier if Fig 6 was reformatted so that subparts appear in order of the text, when possible.

- L445: do you mean lambda phage integration site?

- In the methods overall, there are a few sentences with the verb missing, for instance L486-487, or line 590-591 "this function ?? the cost".

- L494: growth rate estimates of strains growing separately are not competitive fitness.

- In terms of the model, I believe it would be worth having some more details of the equations used, at least within the supplementary material, for the reader to have more of an idea about it without having to go to the full code itself.

Reviewer #2 (Remarks to the Author):

In this manuscript, Hernandez-Beltran and coauthors provide simulations and high-throughput single-cell experiments supporting the hypothesis that the stochasticity in plasmic numbers may explain transient mechanisms of resistance. The study provides interesting new results to understand how some

cells within a population show transient survival to lethal concentrations based on the proposed mechanism.

In my opinion, the experimental work conducted to test the hypothesis was impressive, and it certainly provides solid evidence. Nevertheless, it is unclear what is the objective of the model. It seems like it was built to guide the experiments, which are the ones testing the hypothesis. Nevertheless, stronger results are obtained when the model and data are combined to test the plausibility of the hypothesis. However, I have encountered some difficulties in determining if the modelling simulations were adequate for this second more interesting objective. See the comments below in this regard:

- The main difficulty is that the model was not described mathematically or with sufficient detail. I understand that including this in multi-disciplinary work is not always the best option (especially for agent-based models), but in the current form of the manuscript, it is difficult to assess the suitability of the model to test the proposed hypothesis. More details should be provided, for example, including a supplementary report with critical equations describing the major model hypothesis. On the other hand, the authors provide the code, that is very helpful and fundamental for reproducibility in agent-based models. Nevertheless, it is difficult to navigate to find the considered assumptions. In other words, the reader should have a more detailed sense of the assumptions in the model without the need for running or reading the code. Therefore I have many different questions that I guess are more a lack of information than a major concern regarding the modelling.

- The model has not been sufficiently confronted with the data, even when very appropriate measurements were available. For example, the model assumes Gaussian (and therefore symmetric) probability distributions for PCN without treatment. The simulated distributions change and are not symmetric during AMP exposure as shown in Figures 1E and S1. However, these simulations were not compared with the distributions measured and depicted in Figure 2, just showing this asymmetry. Not sure without seeing the results, but this could be more interesting than the comparison made in Figure 6.

- Related to the previous comment, I wonder how valid is to assume Gaussian PCN distributions when there is no treatment. Is this assumption supported by the literature? Have you explored if the modelling results change considerably when based on distributions with a tail (meaning that there are few cells with extremely high PCN)?

-Also related, agent-based models are based on probability distributions that should be well-supported by the literature or clearly stated as considered assumptions. A good approach is to include a table with the different considered distributions, and if there are, experimental or theoretical works supporting the different selections.

- It is unclear what the authors mean by the sentence in line 76 "The propensities of each process are determined from the concentrations of a limiting resource and a bactericidal antibiotic present in a well-mixed environment". Is this meaning that propensities were estimated from obtained data not presented in this work?

- Regarding the shared code, a more detailed readme file is needed, for example, which is the main file to compute the model. It is clear how to depict the data, but not so simple to guess how to simulate the model and generate the data for the figures.

- Probably sentence in line 401 should be relaxed "Population-level experiments cannot distinguish between these hypotheses". Single-cell analysis is more appropriate but does not mean that some information cannot be obtained to discern between both hypotheses. For example, could it be that for some rates of division and death, probably some decrease in the dynamics of the population might be detected for the hypothesis of preexisting cell-to-cell heterogeneity in resistance levels?

- More details, than that provided in line 505, are needed on how the model simulates the action of the antibiotic.

It should be mentioned that my background is in theoretical modelling which explains the focus of most of the comments.

Reviewer #3 (Remarks to the Author):

In this paper, Hernandez-Beltran and colleagues study how bacterial populations evolve plasmid-mediated antibiotic resistance from intercellular heterogeneity in the plasmid copy number. On the one hand, they present results from a computational model of plasmid dynamics, i.e., stochastic replication and segregation of plasmid copies during bacterial growth. On the other hand, they measure the expression level of a β -lactamase-encoding multicopy plasmid in *E. coli* using a plasmid-encoded GFP gene. This experimental approach allows them to indirectly determine the plasmid copy number (PCN) of single cells of populations treated with sub-inhibitory concentrations of a β -lactam antibiotic. Their results show a shift of the PCN distribution that is positively correlated with the applied antibiotic concentration, aligning with their theoretical results. Using population survival assays, they exposed bacterial strains carrying plasmids with different PCN to antibiotics. They found that the median survival time at MIC is substantially higher for bacteria carrying a multicopy plasmid than for the control strain carrying the β -lactamase gene in only one copy in the chromosome. Furthermore, their results show a

positive correlation between the rate of increase of the GFP level (reflecting the copy number) with the increased median survival times, which indicates that the high degree of observed copy number heterogeneity in multicopy plasmids facilitates resistance evolution by a rapid adaptation of the PCN and thus the copy number of the resistance gene. Overall, the manuscript is written very clearly. It presents essential aspects of plasmid copy number variations and plasmid dynamics' influence on antibiotic resistance evolution.

I note that I am not qualified to comment on the experimental side of this study. Hence, I focus my review on the theoretical side of this work.

Major comments:

1A. The origin of PCN heterogeneity in the computational model.

I am concerned by the mathematical/computational model setup for the plasmid dynamics. The authors state that each cell has a specific (maximum) plasmid copy number derived from a Normal distribution $N(\mu, \sigma)$. Hence, the model assumes a priori the heterogeneity of the plasmid copy number (by unequal plasmid control systems of different cells). Yet, the heterogeneity of the PCN could be entirely the consequence of several other factors of plasmid dynamics, e.g., stochastic plasmid replication, uneven plasmid segregation, and selection for plasmid-encoded genes, which are additionally incorporated in the model. Other factors, such as post-segregational-killing systems that eliminate plasmid-free cells, might also play a role in maintaining the PCN heterogeneity. It needs to be clarified if the PCN heterogeneity results from the maximal PCN distribution or from the stochastic plasmid dynamics.

1B. Parameters of the computational model.

The coefficient of variation of the maximal PCN (σ) is a crucial computational model parameter. To my understanding of the study, this parameter σ and several other model parameters were set without experimental validation. It would be interesting to estimate the PCN number variation of pBGT from the GFP intensities to provide proof for the chosen parameter of 15% (10%, see minor comment on Table S2) for the computer simulations.

2. Antibiotic concentrations in the population-level survival assay.

To determine the median survival time for the control strain (MG:GT, without plasmid), an antibiotic concentration of 4xMIC (0.5mg/ml) was used. However, for the plasmid-carrying strain (MG/pBGT), the MIC (32mg/ml) was used. Therefore, it is arguable if the higher median survival time for MG/pBGT compared to the control is due to the phenotype noise (arising from PCN variance) or the differences in antibiotic stress.

Minor comments:

– L27: Consider replacing "provide further support to" with "further support the."

- L73: Add a comma after "first".
- L82: To my understanding of the presented model and the supplied simulation code,
- L84: It seems that the PCN heterogeneity is more an assumption of the model rather than a "consequence" or "result" (see L86) of the stochastic plasmid dynamics (see comment 1.)
- Figure 1: Please streamline the x-axis in plots A and D and denote that the time is in units of minutes in the methods section.
- Figure 1B: Is this the distribution of the current PCN or the maximum PCN? If these are the maximum plasmid copy numbers, please clarify that in the figure caption. What is the time point if these are the current copy numbers?
- Figure 1D: What is the black line showing? What is the difference between the black and the red lines? Please clarify in the caption.
- Figure 1F: There is no dotted line visible.
- L108: "A)" missing
- L122: Please add "[cells that were killed are marked] with a red dot" (see comment above)
- L125: There is no black line.
- L145: Does this apply to protein expression in general or to the specific expression of β -lactam genes? More a question out of curiosity: It is unclear if the β -lactamase production, thus, depends linearly on the gene copy number. Could it be that, for example, capacity restrictions in the translation machinery result in a saturation of the beta-lactamase production and, thus, in a saturation of the MIC for higher copy numbers (see e.g. the similar MICs of strains carrying the plasmids MG/G54U and MG/G55U, where the latter has a PCN that is two-fold higher)
- L153: Remove comma.
- L154: Why is the Fold change in PCN only shown for the plasmids MG/G54U and MG/G55U (Figure S3) and not for the pBGT plasmid? Couldn't this be done by the regression results shown in Figure S2?
- L158: I find it confusing that for pBGT, only a result is shown for the GFP, and for MG/G54U and MG/G55U for the PCN fold change.
- L160: Include "Fig 2E" in the brackets. You might consider moving this sentence to L156 to compare pBGT and MG:GT more directly.
- Figure 2C: Something is peculiar with the x-axis ranges of Figure 2C.
- Figure 2E: It should be indicated that there is a jump on the x-axis between 0 and positive values shown on a logarithmic scale. What do the dotted lines show? If it results from a simple non-linear regression, the value for zero should not be included.
- L176: "a sample". Is a sample taken from each of the 88 populations?

- Figure 3C: In L520, it says that samples were taken every 30min, from 0-8 hours, then at 18h and 24h. This is inconsistent with the times shown in Fig 3C. Were the sample time points different? Isn't this the same experiment?
- Figure 3A: If I understand correctly from the method described in L514, 88 populations were continuously growing on an LB+AMP plate with samples taken every 30 minutes for the first 8h and after 18h and 24h. So a small test volume from the original LB+AMP plate was injected into an LB plate. The illustration in Fig 3A is misleading. In each row, there are 8 wells of the 96 well plates highlighted in red, and I do not see how this relates to the experimental setup as there are not several LB+AMP well plates with bacteria growing for different durations but only single LB+AMP well plates and samples were taken at different times.
- L191: Add Fig 3C in brackets.
- L220: Why would you not expect an increase in the mean cell-individual GFP when low-copy-number cells are eliminated, as shown for the population level?
- L233: I find the reference to Figure 4D confusing, as I cannot see the difference in cell division rates between surviving and non-surviving cell lineages. Do the colors (orange/blue) have the same meaning as in Figure 4B? Otherwise, please consider changing the colors.
- L239 "54% survival for the top quartile [...] 32%" for the bottom quartile: However, why does the graph in Figure 4F show a survival rate of ~75% for the ">74" group (top quartile)? Why do you show the Normalized GFP intensity only for values down to 0.32? Wouldn't including <0.25 into the plot make sense to show the 32% survival?
- Figure 4D: The x-axis label is 'Number of divisions'. How can these values be non-integer values?
- L296: Please include a reference to Figure 4B here.
- L300: I do not understand how to read Figure S11 to obtain the statement made in L299-300. However, can't you immediately obtain the two distinct characteristics from Figure 4C: Survival of cells that did not filament needed a higher GFP than the survival of cells that filamented (see the shift of the dark blue line compared to the light blue line)?
- Figure 5C plot 3: Why are there no dead cells shown in the third plot while the population fraction of dead cells is approximately 15-20% in Figure 5D?
- L341 "plasmid replication and segregation of plasmids ensure that the whole PCN distribution is reconstituted by any cell". I do not understand this sentence, especially what you mean by "by any cell". I wondered if you meant that the replication and segregation ensure the original PCN distribution is restored.
- L345 "strength of selection [for lower PCN]": Please clarify that you are referring to selection for lower PCN.
- L354: Please check the order of the supplemental figures.
-
-

- L588: It is unclear if plasmid copy numbers are derived from the Normal distribution only initially or over the entire course of the simulation. Please clarify.
- In L589, you call μ the mean copy number while it is the max PCN in Table S2.
- L590: Verb missing: "This function the cost entailed".
- L589: Are cells characterized by size and energy (ATP concentration)? Please clarify in the text.
- L593: Please modify "They [Plasmids] began to replicate plasmids" -> "Plasmids are replicated following a probability"
- L594: How often do plasmids replicate per generation? I quickly looked in the code and saw a parameter set to 3. Is that the maximum number of replications per minute? If yes, it is still unclear why a maximum number of three replications per minute was chosen.
- L595: Please give more information on the individual resistance profile and how the 'action of the antibiotic' is implemented. There should be an expression for how the antibiotic concentration alters the probability of survival and how 'random noise' affects the survival chance.
- Figure S1, PCN legend: Why is there a bin for PCN=0 (grey) if there is already a bin for PCN=0.1 (very light green)?
- Figure S1B, caption, "plasmid-free in white": Please replace "white" with grey (see also comment above).
- Figure S2: A and B missing.
- Figure S2: What is the difference between the green dots and the green star (near label "R4")?
- Figure S2B: Does the SSC (y-axis) indicate the cell size here? How is it related to the PCN?
- Figure S2: A) What type of regression was used here? Please provide the underlying regression function that was used. Has the result of the regression been used anywhere else?
- Figure S2: B) Why are there four regions R1-R4? However, only three regions (low, medium, and high) are mentioned in the text.
- L797: What antibiotic concentration was used here?
- Figure S12: I am not an expert in PCA analysis, but if I understand the plot correctly, I obtain that cells (both filamented and non-filamented) have a smaller gfp value if they survived the treatment (see smaller values for PC2 loaded with both gfp_first and gfp_last) which would be contrary to your conclusions.
- Table S2: Please indicate how the parameters (except for the copy number) were chosen and which parameters are calibrated to the experimental setup. Why is $\sigma=0.1$ here but $\sigma=0.15$ in Figure 1B (black line)

Reviewer #1 (Remarks to the Author)

This is a very high-quality and detailed paper, that addresses the consequences of plasmid copy number variability in multicopy plasmids on population-level heterogeneity and response to antibiotic selection. The authors have previously studied in detail other aspects of copy number dynamics; here they focus on fluctuating environments. The results are nicely discussed in the context of both plasmid biology and phenotypic heterogeneity.

Thank you for the positive feedback and for recognizing the depth of our work. We value the appreciation of our efforts in studying the dynamics of plasmid copy number and are pleased that our discussion resonated in the broader context of plasmid biology and phenotypic heterogeneity.

I have a few comments below, overall minor, in order of appearance in the text:

- L63 'an adaptive strategy' – for whom? Which level of selection is this?

A clarification has been incorporated in the manuscript.

- L67 encoding genes is beneficial

Changed. Thanks.

- L71 and elsewhere: maybe avoid the use of the word 'tolerance' in this manuscript, as this is used to describe mechanisms that do not involve changes in antibiotic resistance itself, whereas the phenomenon described here fully relies on resistance gene expression.

The reviewer is correct. We have refrained from using the word 'tolerance' in this context.

- Figure 1: "A" is lacking from the legend text; the gradient of greens in 1A could be added as a color legend to make more obvious which side of the gradient corresponds to high PCN. In 1F, I can see a red line, but no black line and no dotted line described in the legend.

We appreciate the reviewer's keen observation. We have updated Figure 1 to address the discrepancies. The gradient of greens in 1A has been supplemented with a color legend for clearer representation of PCN values. Additionally, we have corrected the visual elements in 1F to match the description in the legend.

- Figure 3: could the authors explain, or comment on why there is no difference in survival between pBGT and G54U populations despite the higher copy number of G54U?

We appreciate the query regarding Figure 3. While G54U does exhibit a higher copy number, our observations suggest that it's not just the mean plasmid copy number (PCN) that impacts survival. Instead, our data indicates that the coefficient of variation in PCN might play a more influential role in determining survival probability. This underscores the complexity and nuances of how PCN dynamics can influence cellular responses to antibiotics.

- Movies S1, S2 and Fig 5C: why does one of these populations have no red (dead) cells at all, if the semi-lethal antibiotic concentration has been normalised so that each strain experiences the same amount of death?

We understand the concerns raised regarding the depiction of dead cells in Movies S1, S2, and Fig 5C. In our image analysis pipeline and experimental setup, dead cells are only discernible for a limited number of frames. As these cells deteriorate and lose their structural integrity, the rhodamine signal weakens and eventually isn't captured in the subsequent frames.

Furthermore, we would like to highlight that our visual data comes from an assortment of microfluidic chambers. While we aim for consistency, there is inherent variability in individual chambers, and the particular representation in one movie or figure might not entirely reflect the aggregate cell death across all chambers. We have modified the corresponding Figure to avoid confusions.

- Figure 4D shows the number of divisions whereas the text L232 measures the time elapsed between cell division events, using the same metric for both would be easier to follow for the reader. Similarly, Figures 4E and F are divided into 9 bins of GFP intensity but the text mentions quartiles instead.

A correction in the text has been made, accompanied by including a differentiation between Survived and Not-survived cells.

- L304: enabling the population to survive

Corrected.

- L324 and whole paragraph: Fig 5C does not really make the point about cell division time described in the text. I wonder if it wouldn't also be worth introducing plasmid cost (Fig S13) earlier, even within Fig 2 together with Fig 2D, as part of the initial characterisation of the system.

We removed the Figure reference, which indeed was left out from previous versions of the manuscript.

- L346: I did not understand the point the authors are making about decay of selection, or how this is visible in Fig 6D.

We modified the text to make it clear.

- The whole section page 13 of the manuscript is a bit difficult to follow, it feels like subsections of figures have been moved around too many times, and it would be easier if Fig 6 was reformatted so that subparts appear in order of the text, when possible.

We sincerely appreciate the reviewer's feedback regarding the manuscript's organization. Upon review, we concur that the structure was not optimal. To address this, we have restructured this section and integrated the results into more appropriate areas of the manuscript, ensuring a more logical flow. We believe this modification improves the readability and coherence of the presentation

- L445: do you mean lambda phage integration site?

Yes, we have corrected this.

- In the methods overall, there are a few sentences with the verb missing, for instance, L486-487, or line 590-591 "this function ?? the cost".

Thanks for pointing this out.

- L494: growth rate estimates of strains growing separately are not competitive fitness.

Corrected

- In terms of the model, I believe it would be worth having some more details of the equations used, at least within the supplementary material, for the reader to have more of an idea about it without having to go to the full code itself.

Thank you for your valuable suggestion regarding the inclusion of the model's equations. We have now added a detailed description of the model in the supplementary material to aid readers in understanding without needing to refer to the full code. The code used to generate the supplementary material and the manuscript's figures is available as a series of Jupyter Notebooks deposited in a public repository: <https://github.com/ccg-esb-lab/pBGT>

Reviewer #2 (Remarks to the Author):

In this manuscript, Hernandez-Beltran and coauthors provide simulations and high-throughput single-cell experiments supporting the hypothesis that the stochasticity in plasmic numbers may explain transient mechanisms of resistance. The study provides interesting new results to understand how some cells within a population show transient survival to lethal concentrations based on the proposed mechanism.

In my opinion, the experimental work conducted to test the hypothesis was impressive, and it certainly provides solid evidence. Nevertheless, it is unclear what is the objective of the model. It seems like it was built to guide the experiments, which are the ones testing the hypothesis. Nevertheless, stronger results are obtained when the model and data are combined to test the plausibility of the hypothesis. However, I have encountered some difficulties in determining if the modelling simulations were adequate for this second more interesting objective.

We appreciate the reviewer's positive feedback on our experimental work and recognize the concerns raised regarding the clarity of our model's objective. The primary intention behind the model was indeed to guide our experiments. However, it was also designed to provide a theoretical framework that complements and strengthens the empirical results, allowing for a more profound understanding of the transient resistance mechanisms.

Following the reviewer's suggestion, we have now employed the model to evaluate the robustness of our results. Specifically, we examined various sources of noise and the effects of different plasmid copy number (PCN) distribution shapes. This enhanced application of the model further validates our findings and underscores the importance of our theoretical framework in the broader context of the study.

We also understand the necessity for clear elaboration on how our simulations align with the overarching objectives. Thus, we have expanded on the model's explanation in a new Supplementary Information. This will offer a detailed breakdown of each parameter, its implications, and how they interplay to shape our results.

See the comments below in this regard:

- The main difficulty is that the model was not described mathematically or with sufficient detail. I understand that including this in multi-disciplinary work is not always the best option (especially for agent-based models), but in the current form of the manuscript, it is difficult to assess the suitability of the model to test the proposed hypothesis. More details should be provided, for example, including a supplementary report with critical equations describing the major model hypothesis. On the other hand, the authors provide the code, that is very helpful and fundamental for reproducibility in agent-based models. Nevertheless, it is difficult to navigate to find the considered assumptions. In other words, the reader should have a more detailed sense of the assumptions in the model without the need for running or reading the code. Therefore I have many different questions that I guess are more a lack of information than a major concern regarding the modelling.

Thank you for pointing out the need for a more detailed description of our model. We acknowledge that in our pursuit to maintain the manuscript's conciseness, some important details might have been left out. To rectify this, we have now included supplementary material with a comprehensive explanation of the model. Alongside, the supplementary material is accompanied by Jupyter Notebooks to further aid in understanding (<https://github.com/ccg-esb-lab/pBGT>). We appreciate the reviewer's feedback as it has allowed us to enhance the accessibility of our work.

- The model has not been sufficiently confronted with the data, even when very appropriate measurements were available. For example, the model assumes Gaussian (and therefore symmetric) probability distributions for PCN without treatment. The simulated distributions change and are not symmetric during AMP exposure as shown in Figures 1E and S1. However, these simulations were not compared with the distributions measured and depicted in Figure 2, just showing this asymmetry. Not sure without seeing the results, but this could be more interesting than the comparison made in Figure 6.

We thank the reviewer for drawing attention to the observed asymmetry in the distributions. We have now taken time to discuss this further in our manuscript. The skewness in the distribution, as the reviewer rightly observed, emerges from the antibiotic's impact on cells with low PCN, leading to a shift in the mean PCN towards higher values. This observation is indeed central to our results. The existence of cells with high PCN values (that survive the treatment) in conjunction with those with low PCN values (that recover swiftly post-treatment) is a crucial insight that our study aims to highlight.

- Related to the previous comment, I wonder how valid is to assume Gaussian PCN distributions when there is no treatment. Is this assumption supported by the literature? Have you explored if the modelling results change considerably when based on distributions with a tail (meaning that there are few cells with extremely high PCN)?

In response to the reviewer's suggestion regarding the Gaussian distribution assumption for PCN, we revisited our model. While our initial approach employed a Normal distribution for PCN, we expanded our analysis to consider alternative distributions. The results of this deeper exploration are presented in the new supplementary material.

We tested the effects of selection on PCN across multiple distributions, including those with pronounced tails. Consistently, our findings reveal that the specific shape of the distribution has less influence than the magnitude of the variance, $\sigma^2 > 0$. This robust observation, incorporated in our supplementary material, strengthens our conclusions.

-Also related, agent-based models are based on probability distributions that should be well-supported by the literature or clearly stated as considered assumptions. A good approach is to include a table with the different considered distributions, and if there are, experimental or theoretical works supporting the different selections.

Thank you for highlighting the importance of detailing the probability distributions underpinning our agent-based models. In response to your suggestion, we have updated the parameters table to include specific references for those that are grounded in literature. We believe this enhancement will offer readers a clearer understanding of the foundational assumptions of our model.

- It is unclear what the authors mean by the sentence in line 76 "The propensities of each process are determined from the concentrations of a limiting resource and a bactericidal antibiotic present in a well-mixed environment". Is this meaning that propensities were estimated from obtained data not presented in this work?

We thank the reviewer for highlighting the unclear phrasing. Our intention was to convey that cell death and duplication rates are influenced by the availability of resources, while cell death is also contingent on the antibiotic concentration. We understand the original phrasing may have been ambiguous, so we have revised the sentence for clearer communication. We did not infer any estimations from external, non-presented data in this work

- Regarding the shared code, a more detailed readme file is needed, for example, which is the main file to compute the model. It is clear how to depict the data, but not so simple to guess how to simulate the model and generate the data for the figures.

We appreciate the reviewer's recommendation regarding the shared code. In response, we have included a series of Jupyter Notebooks that describe the model and its parameters, as well as provide numerical examples. These additions are intended to facilitate a more accessible simulation of the model and generation of the figures included in the manuscript and the supplementary material.

- Probably sentence in line 401 should be relaxed "Population-level experiments cannot distinguish between these hypotheses". Single-cell analysis is more appropriate but does not mean that some information cannot be obtained to discern between both hypotheses. For example, could it be that for some rates of division and death, probably some decrease in the dynamics of the population might be detected for the hypothesis of preexisting cell-to-cell heterogeneity in resistance levels?

We thank the reviewer for the thoughtful observation. While we acknowledge that certain conditions in population-level experiments may reveal features in their dynamics suggesting preexisting cell-to-cell heterogeneity in resistance levels, such observations are inherently averaged over heterogeneous populations. In our view, single-cell approaches provide a distinct advantage by allowing for a detailed reconstruction of this heterogeneity, which can potentially offer a more granular understanding of resistance mechanisms. We respect the reviewer's perspective and have strived to clarify our stance in the revised manuscript without dismissing the potential merits of population-level experiments.

- More details, than that provided in line 505, are needed on how the model simulates the action of the antibiotic.

We recognize that our initial description in the manuscript may have been insufficient. In response to the reviewer's suggestion, we have incorporated an explicit explanation of this process in the new supplementary material. This additional detail aims to provide readers with a thorough understanding of the simulation and its underlying mechanisms.

It should be mentioned that my background is in theoretical modelling which explains the focus of most of the comments.

Reviewer #3 (Remarks to the Author)

In this paper, Hernandez-Beltran and colleagues study how bacterial populations evolve plasmid-mediated antibiotic resistance from intercellular heterogeneity in the plasmid copy number. On the one hand, they present results from a computational model of plasmid dynamics, i.e., stochastic replication and segregation of plasmid copies during bacterial growth. On the other hand, they measure the expression level of a β -lactamase-encoding multicopy plasmid in *E. coli* using a plasmid-encoded GFP gene. This experimental approach allows them to indirectly determine the plasmid copy number (PCN) of single cells of populations treated with sub-inhibitory concentrations of a β -lactam antibiotic. Their results show a shift of the PCN distribution that is positively correlated with the applied antibiotic concentration, aligning with their theoretical results. Using population survival assays, they exposed bacterial strains carrying plasmids with different PCN to antibiotics. They found that the median survival time at MIC is substantially higher for bacteria carrying a multicopy plasmid than for the control strain carrying the β -lactamase gene in only one copy in the chromosome. Furthermore, their results show a positive correlation between the rate of increase of the GFP level (reflecting the copy number) with the increased median survival times, which indicates that the high degree of observed copy number heterogeneity in multicopy plasmids facilitates resistance evolution by a rapid adaptation of the PCN and thus the copy number of the resistance gene. Overall, the manuscript is written very clearly. It presents essential aspects of plasmid copy number variations and plasmid dynamics' influence on antibiotic resistance evolution.

We thank the reviewer for stating that is written very clearly and that it represents essential aspects of plasmid dynamics and antibiotic resistance.

I note that I am not qualified to comment on the experimental side of this study. Hence, I focus my review on the theoretical side of this work.

Major comments:

1A. The origin of PCN heterogeneity in the computational model.

I am concerned by the mathematical/computational model setup for the plasmid dynamics. The authors state that each cell has a specific (maximum) plasmid copy number derived from a Normal distribution $N(\mu, \sigma)$. Hence, the model assumes a priori the heterogeneity of the plasmid copy number (by unequal plasmid control systems of different cells). Yet, the heterogeneity of the PCN could be entirely the consequence of several other factors of plasmid dynamics, e.g., stochastic plasmid replication, uneven plasmid segregation, and selection for plasmid-encoded genes, which are additionally incorporated in the model. Other factors, such as post-segregational-killing systems that eliminate plasmid-free cells, might also play a role in maintaining the PCN heterogeneity. It needs to be clarified if the PCN heterogeneity results from the maximal PCN distribution or from the stochastic plasmid dynamics.

We are grateful for the insightful observation made by the reviewer regarding the origin of PCN heterogeneity in our computational model. Our model inherently introduces PCN heterogeneity by setting a maximal plasmid copy number derived from a Normal distribution, capturing the baseline variability. Furthermore, our model embraces other external contributors to PCN dynamics such as stochastic plasmid replication and uneven plasmid segregation during division. This modeling approach mirrors the observed natural variability in bacterial populations caused by imperfect plasmid control across individual cells. Importantly, our framework allows for the exploration of the cumulative effects of these diverse noise sources on PCN variations, giving insights into how variations in PCN distribution influence bacterial survival in various antibiotic environments.

We appreciate the reviewer's emphasis on the sources of noise and their contributions to PCN heterogeneity. In response to this, we carried out additional computational experiments isolating different noise sources and observing the PCN distribution over time. Our findings indicate that the specific source of noise is less critical for bacterial survival. Instead, what emerges as a pivotal factor is the resulting distribution's variance. Regardless of the distinct noise sources, a large variance in the PCN distribution consistently proved influential for survival. This insight underscores our study's focus on the population-level implications of cell-to-cell variability.

We appreciate the mention of post-segregational-killing systems and their potential role in maintaining PCN heterogeneity. However, it's crucial to specify that such dynamics are associated with other plasmid types and are not encompassed within the scope of our current model or experimental system. In our study, the inherent PCN heterogeneity is introduced through the stochastic nature of replication and segregation of multicopy plasmids. We have modified the discussion to make this clarification.

1B. Parameters of the computational model.

The coefficient of variation of the maximal PCN (σ) is a crucial computational model parameter. To my understanding of the study, this parameter σ and several other model parameters were set without experimental validation. It would be interesting to estimate the PCN number variation of pBGT from the GFP intensities to provide proof for the chosen parameter of 15% (10%, see minor comment on Table S2) for the computer simulations.

We thank the reviewer for highlighting the significance of the coefficient of variation of the maximal PCN in our computational model. In response, we have conducted measurements of this parameter, and

these results are now presented in Figure 2. Through these measurements, we observed that a higher CV leads to a pronounced shift in the mean PCN during antibiotic exposure. In our simulations, we have decided to work with smaller CVs based on these findings.

2. Antibiotic concentrations in the population-level survival assay.

To determine the median survival time for the control strain (MG:GT, without plasmid), an antibiotic concentration of 4xMIC (0.5mg/ml) was used. However, for the plasmid-carrying strain (MG/pBGT), the MIC (32mg/ml) was used. Therefore, it is arguable if the higher median survival time for MG/pBGT compared to the control is due to the phenotype noise (arising from PCN variance) or the differences in antibiotic stress.

We thank the reviewer for raising this crucial point regarding the potential differences in antibiotic stress between the control strain (MG:GT, without plasmid) and the plasmid-carrying strain (MG/pBGT). We'd like to clarify that while the absolute antibiotic dose used for pBGT (32mg/ml) appears distinct, it is in fact equivalent to 4x the reported MIC (8 mg/ml) for this strain. This 4xMIC value is determined from consistent measurements taken across the various laboratories involved in our study. It's important to note that we always conduct standard dose-response curves before each experiment to determine the MIC, ensuring that the comparisons between strains are done under equivalent stress conditions.

Minor comments:

– L27: Consider replacing "provide further support to" with "further support the."

Done.

– L73: Add a comma after "first".

Corrected

– L82: To my understanding of the presented model and the supplied simulation code,

This comment appears to be incomplete.

– L84: It seems that the PCN heterogeneity is more an assumption of the model rather than a "consequence" or "result" (see L86) of the stochastic plasmid dynamics (see comment 1.)
reply{We have addressed this comment. In short, yes, it is a model assumption but the overall time-resolve PCN variability emerges from the joint effects of cell-determined maximal copy-number and the stochasticity of the plasmids dynamics.

We appreciate the reviewer's observation on the source of the PCN heterogeneity in our model. In response, we would like to argue that the inherent PCN heterogeneity isn't merely a consequence, but a deliberate design choice. By intentionally embedding this heterogeneity, we're able to control and vary it, allowing us to systematically evaluate how distributions of differing shapes and variances correlate with survival. This approach serves not only to mirror the natural variability observed but also provides a strategic framework for probing specific bacterial dynamics in a controlled manner.

– Figure 1: Please streamline the x-axis in plots A and D and denote that the time is in units of minutes in the methods section.

We thank the reviewer for the suggestion. We have revised Figure 1 and streamlined the x-axis in plots A and D to ensure clarity. Additionally, we have updated the methods section to explicitly mention that the time is denoted in units of minutes. Furthermore, as per the recommendation, a concise model description has been added to the main manuscript and a more detailed explanation is now available in the supplementary material.

– Figure 1B: Is this the distribution of the current PCN or the maximum PCN? If these are the maximum plasmid copy numbers, please clarify that in the figure caption. What is the time point if these are the current copy numbers?

We thank the reviewer for bringing attention to this detail in Figure 1B. We have now clarified in the revised manuscript that the distributions displayed represent the maximum plasmid copy numbers. The necessary updates to the figure caption have been made to ensure this information is explicitly mentioned.

– Figure 1D: What is the black line showing? What is the difference between the black and the red lines? Please clarify in the caption.

We have updated the figure caption to better delineate the distinction between the black and red lines and their respective significance.

– Figure 1F: There is no dotted line visible.

Figure updated.

– L108: "A)" missing

Corrected

– L122: Please add "[cells that were killed are marked] with a red dot" (see comment above)

We have incorporated this suggestion.

– L125: There is no black line.

We have corrected the figure.

– L145: Does this apply to protein expression in general or to the specific expression of β -lactam genes? More a question out of curiosity: It is unclear if the β -lactamase production, thus, depends linearly on the gene copy number. Could it be that, for example, capacity restrictions in the translation machinery result in a saturation of the β -lactamase production and, thus, in a saturation of the MIC for higher copy numbers (see e.g. the similar MICs of strains carrying the plasmids MG/G54U and MG/G54U, where the latter has a PCN that is two-fold higher)

We thank the reviewer for this insightful observation. It's true that saturation phenomena can arise due to multiple reasons, as rightfully indicated by the reviewer. The nature of the resistance protein being expressed can greatly influence this. For example, even a single copy of the CAT gene, which confers resistance to chloramphenicol, can result in considerable resistance levels, making it somewhat decoupled from a strict PCN-activity correlation. However, for the beta-lactamase in our study, and as supported by other research findings, a direct correlation between PCN and activity tends to hold. We have also further elaborated on this in our manuscript to provide clarity.

– L153: Remove comma.

Removed

– L154: Why is the Fold change in PCN only shown for the plasmids MG/G54U and MG/G55U (Figure S3) and not for the pBGT plasmid? Couldn't this be done by the regression results shown in Figure S2?

We conducted the single-cell measurements on multi-dose antibiotics exclusively for the strains MG:GT and MG/pBGT. Although it is technically possible to derive PCN fold change for pBGT using the regression from Figure S2, we refrained from doing so. The measurements for Figure S2 were acquired using a different instrument, making direct comparisons potentially misleading. Therefore, we chose to avoid such a comparison to ensure the accuracy and reliability of our findings.

– L160: Include "Fig 2E" in the brackets. You might consider moving this sentence to L156 to compare pBGT and MG:GT more directly.

Thanks for the suggestion.

– Figure 2C: Something is peculiar with the x-axis ranges of Figure 2C.

The appropriate axis labels have been changed.

– Figure 2E: It should be indicated that there is a jump on the x-axis between 0 and positive values shown on a logarithmic scale. What do the dotted lines show? If it results from a simple non-linear regression, the value for zero should not be included.

The point raised regarding the x-axis and the regression in Figure 2E is valid. An oversight occurred in the initial regression process. The figure has been updated to reflect a linear regression using the actual antibiotic doses to rectify this error.

– L176: "a sample". Is a sample taken from each of the 88 populations?

Yes, we have specified the sampling process in the text.

– Figure 3C: In L520, it says that samples were taken every 30min, from 0-8 hours, then at 18h and 24h. This is inconsistent with the times shown in Fig 3C. Were the sample time points different? Isn't this the same experiment?

We apologize for the oversight. The data and time points described in L520 indeed correspond to Fig 3B. This figure, based on an independent experiment, displays the growth dynamics in AMP. The inconsistency has been rectified in the manuscript to ensure clarity.

– Figure 3A: If I understand correctly from the method described in L514, 88 populations were continuously growing on an LB+AMP plate with samples taken every 30 minutes for the first 8h and after 18h and 24h. So a small test volume from the original LB+AMP plate was injected into an LB plate. The illustration in Fig 3A is misleading. In each row, there are 8 wells of the 96 well plates highlighted in red, and I do not see how this relates to the experimental setup as there are not several LB+AMP well plates with bacteria growing for different durations but only single LB+AMP well plates and samples were taken at different times.

We acknowledge the potential confusion caused by the illustration in Fig 3A. To avoid misinterpretation, we have decided to remove this diagram for clarity.

– L191: Add Fig 3C in brackets.

Done.

– L220: Why would you not expect an increase in the mean cell-individual GFP when low-copy-number cells are eliminated, as shown for the population level?

We appreciate the reviewer's observation. Indeed, eliminating low-copy-number cells would lead to an increase in overall GFP intensity at the population level. However, when observing the GFP intensity of an individual cell over time, an increase could be attributed to factors other than just plasmid copy number, such as heightened metabolic activity induced by the drugs or potential off-target effects. To address this, we examined these possibilities and our analysis, presented in Figure S16, supports the original statement.

– L233: I find the reference to Figure 4D confusing, as I cannot see the difference in cell division rates between surviving and non-surviving cell lineages. Do the colors (orange/blue) have the same meaning as in Figure 4B? Otherwise, please consider changing the colors.

We have modified the figure to avoid any confusion.

– L239 "54% survival for the top quartile [...] 32%" for the bottom quartile: However, why does the graph in Figure 4F show a survival rate of ~75% for the ">74" group (top quartile)? Why do you show the Normalized GFP intensity only for values down to 0.32? Wouldn't including <0.25 into the plot make sense to show the 32% survival?

Thank you for clarifying the issue. We apologize for the confusion caused by the binning in Figure 4F. We have now adjusted the binning to accurately represent the survival rates in the different quartiles and to include the appropriate range of Normalized GFP intensity values.

– Figure 4D: The x-axis label is 'Number of divisions'. How can these values be non-integer values?

The non-integer values on the x-axis arise because the histogram represents density rather than direct counts. We acknowledge the oversight in the y-axis label, which was omitted. This has now been rectified to enhance clarity.

– L296: Please include a reference to Figure 4B here.

Included.

– L300: I do not understand how to read Figure S11 to obtain the statement made in L299-300. However, can't you immediately obtain the two distinct characteristics from Figure 4C: Survival of cells that did not filament needed a higher GFP than the survival of cells that filamented (see the shift of the dark blue line compared to the light blue line)?

We appreciate the suggestion. Indeed, Figure 4C directly illustrates the point, showing the survival disparity between non-filamented cells with higher GFP intensity and filamented ones. We've now referenced this in the main text. Figure S11 was aimed at providing a deeper dive into the data; however, we recognize its potential for confusion and will work on improving its clarity.

– Figure 5C plot 3: Why are there no dead cells shown in the third plot while the population fraction of dead cells is approximately 15-20% in Figure 5D?

We apologize for the oversight. The discrepancy was due to an error in image processing. We've since updated Figure 5C to better represent the data and avoid confusion. In our image analysis pipeline and experimental setup, dead cells are captured only for a few frames. As cells lose their structural integrity, the rhodamine signal dissipates and is no longer registered in subsequent frames.

Additionally, it's important to note that our dataset is derived from several microfluidic chambers. Although we strive for accurate representation, the selected image might not perfectly capture the overall cell death rate.

– L341 "plasmid replication and segregation of plasmids ensure that the whole PCN distribution is reconstituted by any cell". I do not understand this sentence, especially what you mean by "by any cell". I wondered if you meant that the replication and segregation ensure the original PCN distribution is restored.

We have modified the sentence to make it clearer.

– L345 "strength of selection [for lower PCN]": Please clarify that you are referring to selection for lower PCN.

We appreciate the feedback. The phrasing was ambiguous. By "strength of selection", we're actually referring to the antibiotic's selective pressure, which effectively selects for higher PCNs, not lower ones. Post-antibiotic "decay" in PCN is an outcome of the growth phase of the bacterial population. When the surviving population size is large, it rapidly progresses to the stationary phase, causing the mean copy-number to revert back to its original value. However, during the exponential growth phase, the average PCN can be skewed by newly divided cells that haven't yet achieved their peak copy-number. We have clarified this in the manuscript.

– L354: Please check the order of the supplemental figures.

Checked.

– L588: It is unclear if plasmid copy numbers are derived from the Normal distribution only initially or over the entire course of the simulation. Please clarify.

A clarification has been made both in the short description, as well as in the supplementary material.

– In L589, you call μ the mean copy number while it is the max PCN in Table S2.

A full correction of table S2 has been made.

– L590: Verb missing: "This function the cost entailed".

This subsection has been completely rewritten

– L589: Are cells characterized by size and energy (ATP concentration)? Please clarify in the text.

We have clarified this in the manuscript and in the supplementary information.

– L593: Please modify "They [Plasmids] began to replicate plasmids" -> "Plasmids are replicated following a probability"

Done. Thanks.

– L594: How often do plasmids replicate per generation? I quickly looked in the code and saw a parameter set to 3. Is that the maximum number of replications per minute? If yes, it is still unclear why a maximum number of three replications per minute was chosen.

Thank you for drawing attention to this aspect. The parameter set to 3 in the code represents the upper limit on the number of replications per minute. This specific choice was informed by our observation and literature that indicated a dynamic range for plasmid replication rates. However, it's essential to understand that the actual replication probability follows an asymptotic function. With approximately 30 opportunities for replication prior to cell division, many cells might undergo division without reaching their maximum copy-number. This recurrent process results in the emergence of plasmid-free cells, thereby driving down the average copy-number across the population. We incorporated this mechanism into our model to better capture the biological nuances of plasmid replication within bacterial populations.

– L595: Please give more information on the individual resistance profile and how the 'action of the antibiotic' is implemented. There should be an expression for how the antibiotic concentration alters the probability of survival and how 'random noise' affects the survival chance.

The new supplementary material has incorporated a full detailed description of this process.

- Figure S1, PCN legend: Why is there a bin for PCN=0 (grey) if there is already a bin for PCN=0.1 (very light green)?

The legend has been corrected.

- Figure S1B, caption, "plasmid-free in white": Please replace "white" with grey (see also comment above).

Corrected.

- Figure S2: A and B missing.

Subfigures indicators have been included.

- Figure S2: What is the difference between the green dots and the green star (near label "R4")?

The green star near the label "R4" in Figure S2 was intended to represent the mean of the data points. We recognize that this was not clear, and we have since revised the figure to enhance clarity and avoid any confusion.

- Figure S2B: Does the SSC (y-axis) indicate the cell size here? How is it related to the PCN?

In Figure S2B, the SSC (Side Scatter) on the y-axis does not directly indicate cell size. In flow cytometry, while the Forward Scatter (FSC) correlates with cell size, the Side Scatter (SSC) is more indicative of cell complexity or granularity, which could relate to the shape of the cell. While not directly addressed here, it's worth noting that a larger cell size might indirectly suggest a higher PCN due to an increased intracellular volume or "carrying capacity".

- Figure S2: A) What type of regression was used here? Please provide the underlying regression function that was used. Has the result of the regression been used anywhere else?

In Figure S2A, we utilized a linear regression. We've updated the figure caption to specify this. The results of this regression were specific to the context of sorting and were not applied elsewhere in the manuscript, as this particular instrument for GFP measurement was solely used for that purpose.

- Figure S2: B) Why are there four regions R1-R4? However, only three regions (low, medium, and high) are mentioned in the text.

In Figure S2B, the four regions R1-R4 were demarcated based on the GFP intensity ranges observed. However, we opted not to include R2 in our analysis due to its broad range of GFP intensities, especially considering the logarithmic scale. We have ensured that the text now accurately reflects this decision.

- L797: What antibiotic concentration was used here?

The caption has been changed to say that we used their respective MICs: 43 and 46 mg/ml.

– Figure S12: I am not an expert in PCA analysis, but if I understand the plot correctly, I obtain that cells (both filamented and non-filamented) have a smaller gfp value if they survived the treatment (see smaller values for PC2 loaded with both gfp_first and gfp_last) which would be contrary to your conclusions.

We thank the reviewer the opportunity to clarify this. PCA is designed to reduce the dimensionality of data by representing the variance through principal components. The principal components capture the variance in the data, and the loadings indicate how each variable contributes to that component. The strong positive correlation, indicated by the dark green color for gfp_first/gfp_last with PC2, reflects the variance these variables contribute to PC2 rather than their absolute values.

Regarding the relationship between variables and cell classification, the PCA serves as a tool to visualize the multidimensional data and its inherent structure. For a direct exploration of the relationship mentioned by the reviewer, Figure S11 would be more appropriate and provides a clearer depiction of the subject.

– Table S2: Please indicate how the parameters (except for the copy number) were chosen and which parameters are calibrated to the experimental setup. Why is $\sigma=0.1$ here but $\sigma=0.15$ in Figure 1B (black line)

We appreciate the observation. To address this and provide clarity, we've added supplementary material detailing the rationale for our parameter choices. Within this addition, we've elucidated on the reasons behind each parameter's selection and have given numerical examples that demonstrate their impact on population dynamics. The discrepancy between sigma values arises from specific considerations and constraints for each context, which are also explained in the supplementary material.

Additional author comments

- Table S1 the names of the last two strains were misplaced.
- Figure S13 the PNCs of the last two strains were corrected.
- The caption in Figure S3 was referring to GFP intensity, not to PCN.

REVIEWER COMMENTS

Reviewer #2 (Remarks to the Author):

The modelling methodology has improved considerably thanks to the added supplementary Information. A table including the used models and assumptions, supporting refs and used parameters, is still missing (which could have been useful to get the whole picture). Nevertheless, the authors have made a huge effort to explain most of the relevant parts of the model which are much more transparent now.

However, the work still needs some clarifications (especially adding more connections between the assumptions and available models in the literature) and changes (especially the use of standard log10 transformation for dynamics of population numbers).

Specific comments regarding the manuscript.

- Provide the source of information (even if it was an assumed value based on authors' experience, understandable as models of plasmid dynamics are scarce) for Table S2. This is critical to understand how strong are the considered assumptions and allow future reuse of these parameters or use/extension of the model by future works.

Specific comments regarding the supplementary info. The new supplementary material still needs, however, some improvements:

Page 1

- Why 19 plasmids per cell is assumed a typical number?
- Could the authors provide more details on where in reference 6 it is stated that the resistance is linear with respect to PCN? (Shao, B. et. al (2021). Single-cell measurement of plasmid copy number and promoter activity. Nature Communications, 12(1), 1-9)

Page 3

- Effi is not defined.
- Cannot find the Pcost_per-plasmid in reference 12, could the authors provide the page?

- Plasmid copy number (μ_i) is defined later, but not here when it is the first time appearing
- Regarding the same model, it might help to see a 3D figure showing how ATP changes (z-axis) with resources and plasmid copy numbers (x and y-axis), to illustrate better the behaviour of the ATPi function.
- Figure 2 labels in the caption seem to be wrong.
- Regarding cell division, there are many strong theoretical works discussing and implementing different models for individual cellular division in bacteria, for example, standard adder/sizer/timer in cell physiology or the Cooper–Helmstetter model based on chromosome replication. Nevertheless, in this work, the division rule is based on ATP dynamics without any reference to previous works using same approach. Authors should discuss why a new model is proposed here when there are many alternatives in the literature.

Page 4

- Usual growth bacterial curves are in $\log_{10}(\text{number of cells})$.
- Here for example lag-phase seems to be missing (although difficult to see without the log transformation), something that should be discussed, especially when simulating serial transfer experiments.
- The concept of the maximal copy number and replication delay parameter seems to be included based on the author's expertise. Is there any mathematical model considering individual cell plasmid dynamics in the literature? How the new model relates with population models of plasmid dynamics such as Fedorec, Alex JH, et al. "Two new plasmid post-segregational killing mechanisms for the implementation of synthetic gene networks in Escherichia coli." *Iscience* 14 (2019): 323-334?

Page 5

- Labels are missing in Fig. 5
- Not clear how the parametrization was done, for example: which data is used for this? Which is the source for the 3 points used in the regression? is there not any more useful data in the literature? a linear regression with 3 points is too poor to infer conclusions.
- A linear correlation between PCN and resistance is attributed to ref 6 on page 1 but not discussed here.

Page 6

- IC was not defined in the text (inhibitory concentration?), again number of bacteria should be represented in \log_{10} to allow comparison with usual time-kill curves (and see if there are tails or shoulders...)

- More details on how the model mimics starvation could be interesting, is this starvation assumed before transfer to new media? depends on the number of cells or it is just decreasing the disponibility of ATP?

Page 7

- Mean PCN increases from 19 to 20 during exposure, Is only one plasmid on average so relevant in terms of population dynamics? See the next point related to this.
- Fig. 7 growth curves should be in log10. This is the standard for many different reasons (additive noise, the many differences in order of magnitude...). The relevance is to see if there are changes in the orders of magnitude of the population, but not so relevant tiny changes for maximum numbers.

Page 8

- It is not clear reading the text the relationship between cell division, ATP threshold and antibiotic action. Is this relationship related to the equation in pag 5 such as $A/(m*\mu+b) > \text{antibiotic_action} + N(0,0.1)$? but the antibiotic concentration is already considered in this equation. I think that the text is not clear enough to infer the meaning and implementation of the parameter antibiotic_action. Note that the text states that the antibiotic action is related to ATP and cell division, but later states that shapes plasmid distribution (is this a consequence or the influence of the parameter in plasmid distribution is included explicitly in the model?).
- The replication delay concept is neither clear to me.

Page 9

- Not clear the change in dynamics at time 30, that induces the variability in plasmid distribution within the population.
- Are the modelled dynamics of ATP equal for all the cases in Fig 9?

Page 10

- There is a major issue here in the text stating that it was assumed normal distribution, but data shows a Gaussian trend. What is the difference between a normal distribution and a Gaussian trend? What is a Gaussian trend and which data shows this trend?
- Which antibiotic dynamics are considered to go from the grey distribution to the red one?
- It is interesting that for some distributions the mean of plasmids mean values increases with antibiotics, but not for others. If the antibiotic selects high plasmid copies, it is expected an increase in plasmid mean values for an environment with antibiotic pressure. The authors should clarify why this is not observed in the figures.

Reviewer #3 (Remarks to the Author):

(A formatted version of my comments is attached the pdf file comments.pdf)

The revised version 1 of the manuscript entitled "Plasmid-mediated phenotypic noise leads to transient antibiotic resistance in bacteria" now includes supplemental information that includes a detailed description of the computational model. I have one main comment regarding my previous comments (see 1.). Furthermore, I have to note that some changes need to be added (in the text and some figures), where the rebuttal letter indicates that a change has already been made (see remaining comments below).

The coefficient of variation of the maximal PCN (σ) is a crucial computational model parameter. To my understanding of the study, this parameter σ and several other model parameters were set without experimental validation. It would be interesting to estimate the PCN number variation of pBGT from the GFP intensities to provide proof for the chosen parameter of 15% (10%, see minor comment on Table S2) for the computer simulations.

We thank the reviewer for highlighting the significance of the coefficient of variation of the maximal PCN in our computational model. In response, we have conducted measurements of this parameter, and these results are now presented in Figure 2. Through these measurements, we observed that a higher CV leads to a pronounced shift in the mean PCN during antibiotic exposure. In our simulations, we have decided to work with smaller CVs based on these findings.

1. I recognize that the authors included some numbers in brackets below each colored circle in Fig. 2D; however, I do not find any explanation for these values in the figure caption, which makes the given numbers somewhat useless. (What do the numbers below each circle mean, and if they show the COV, why are they so different from the COV in Fig. 2E?). It is intuitive that a higher CV pronounced shift in the mean PCN. My main issue is that I still cannot understand why the authors chose to work with a small CV of only 10% when experimental measurements show higher variation in the GFP intensity. If a pronounced shift with a higher CV overestimates the change in the copy number, which is why smaller CVs are chosen, it would be necessary to show results from "these [additional] measurements" or simulations.

We thank the reviewer for the suggestion. We have revised Figure 1 and streamlined the x-axis in plots A and D to ensure clarity.

Figure 1D has moved to the SI with no unit in the x-axis label.

L125: There is no black line.

We have corrected the figure.

No, I do not find this corrected (Figure 1E, the black line is maybe below the red shade).

We thank the reviewer for this insightful observation. It's true that saturation phenomena [...] We have also further elaborated on this in our manuscript to provide clarity.

No, I do not find further text in the revised manuscript that provides clarity on the correlation between GFP and PCN.

Figure 2C: Something is peculiar with the x-axis ranges of Figure 2C.

The appropriate axis labels have been changed.

This still needs to be corrected.

Figure 2E: It should be indicated that there is a jump on the x-axis between 0 and positive values shown on a logarithmic scale. What do the dotted lines show? If it results from a simple non-linear regression, the value for zero should not be included.

The point raised regarding the x-axis and the regression in Figure 2E is valid. An oversight occurred in the initial regression process. The figure has been updated to reflect a linear regression using the actual antibiotic doses to rectify this error.

The dotted line cannot result from a linear regression with the actual AB doses, as the linear regression would be a linear function of the form $COV(x)=a+bx$, where x is the actual drug concentration. The function plotted on a linear-log scale (logarithmic x-scale with base 2 excluding $x=0$) would have an exponential form $COV(x)=a+b*2^x$. In other words, the difference $COV(x=1/32)-COV(x=1/64)$ would be 2 times higher as $COV(x=1/64)-COV(x=1/128)$ and so on. See Figure S1, but note that b is positive in Fig S1.

I need help finding Figure S16.

L345 "strength of selection [for lower PCN]": Please clarify that you are referring to selection for lower PCN.

We appreciate the feedback. The phrasing was ambiguous. By "strength of selection", we're actually referring to the antibiotic's selective pressure, which effectively selects for higher PCNs, not lower ones. Post-antibiotic "decay" in PCN is an outcome of the growth phase of the bacterial population. When the surviving population size is large, it rapidly progresses to the stationary phase, causing the mean copy-number to revert back to its original value. However, during the exponential growth phase, the average PCN can be skewed by newly divided cells that haven't yet achieved their peak copy-number. We have clarified this in the manuscript.

I do not find a clarification. The sentence has been removed.

L594: How often do plasmids replicate per generation? I quickly looked in the code and saw a parameter set to 3. Is that the maximum number of replications per minute? If yes, it is still unclear why a maximum number of three replications per minute was chosen.

Thank you for drawing attention to this aspect. The parameter set to 3 in the code represents the upper limit on the number of replications per minute. This specific choice was informed by our observation and literature that indicated a dynamic range for plasmid replication rates. However, it's essential to understand that the actual replication probability follows an asymptotic function. With approximately 30 opportunities for replication prior to cell division, many cells might undergo division without reaching their maximum copy-number. This recurrent process results in the emergence of plasmid-free cells, thereby driving down the average copy-number across the population. We incorporated this mechanism into our model to better capture the biological nuances of plasmid replication within bacterial populations.

The model description should also describe this iterative mechanism of plasmid replication. The 'upper limit on the number of replications per minute' is indeed a model parameter.

Additional comments.

☒ If one looks closely, the tip of the curve in Figure 1D is not visible anymore.

REVIEWER COMMENTS

Reviewer #2 (Remarks to the Author):

The modelling methodology has improved considerably thanks to the added supplementary Information. A table including the used models and assumptions, supporting refs and used parameters, is still missing (which could have been useful to get the whole picture). Nevertheless, the authors have made a huge effort to explain most of the relevant parts of the model which are much more transparent now.

However, the work still needs some clarifications (especially adding more connections between the assumptions and available models in the literature) and changes (especially the use of standard log10 transformation for dynamics of population numbers).

We thank the reviewer for considering that the Supplementary Information has improved the clarity of the model. In accordance with the reviewer's recommendation, we have updated Table S2 to explicitly detail the sources and rationales for the parameter values used in our model. Additionally, we have expanded the Supplementary Information to include comprehensive details on the calibration process for each parameter.

Specific comments regarding the manuscript.

- Provide the source of information (even if it was an assumed value based on authors' experience, understandable as models of plasmid dynamics are scarce) for Table S2. This is critical to understand how strong are the considered assumptions and allow future reuse of these parameters or use/extension of the model by future works.

We acknowledge the reviewer's emphasis on the necessity for transparent and sourced parameters within our model to ensure its utility and reproducibility for future research. Accordingly, we have now included detailed sources for each parameter value used in our simulations, as provided in the updated Table S2. We recognize that while precise data on plasmid dynamics can be elusive, our numerical experiments seem to be robust to specific numerical values, provided that two key conditions are met: a) there exists variability in the copy number of a resistance gene, and b) there is a correlation between the gene dosage and survival probability. These conditions ensure that the natural variability in plasmid replication and segregation suffices to sustain populations with heteroresistance.

Specific comments regarding the supplementary info. The new supplementary material still needs, however, some improvements:

- Why 19 plasmids per cell is assumed a typical number?

This specific value is derived from our reference experimental system (references 24 and 38), where the mean plasmid copy number per cell is observed to be 19.12 ± 1.56 . To ensure the robustness of our numerical results, we subjected our model to simulations across a spectrum of mean PCN values. These additional simulations have demonstrated that the conclusions drawn from our model hold true over a range of PCN averages. The results of these extended analyses are now presented in Supplementary Figure 9.

- Could the authors provide more details on where in reference 6 it is stated that the resistance is linear with respect to PCN? (Shao, B. et. al (2021). Single-cell measurement of plasmid copy number and promoter activity. Nature Communications, 12(1), 1-9)

We are thankful to the reviewer for highlighting this error. In the main text, we have correctly attributed Shao et al. (2021) with findings that indicate a positive correlation between PCN and protein expression. The statement regarding a linear relationship between resistance and PCN was erroneously included in the Supplementary Information. We have revised this to accurately reflect the findings from Shao et al. (2021) and apologise for any confusion caused (Supp Info Lines 32-33).

- Effi is not defined.

The new version of the Supplementary Information now defines this parameter (Supp Info Lines 87-88)

- Cannot find the Pcost_per-plasmid in reference 12, could the authors provide the page?

Thank you for your inquiry regarding the Pcost_per-plasmid parameter. The foundational information for this parameter is detailed on page 2, second paragraph of what is now reference 3 in our revised manuscript. In our study, we have approached this by dividing the total fitness cost, as defined in the reference, by the number of plasmids. We have included an explanation in the Supplementary Information on how this derived parameter is used within our model (Supp Info Lines 96-98).

- Plasmid copy number (μ_i) is defined later, but not here when it is the first time appearing

We have now introduced the definition of plasmid copy number at the first point of use within the Supplementary Information (Supp Info Line 96)

- Regarding the same model, it might help to see a 3D figure showing how ATP changes (z-axis) with resources and plasmid copy numbers (x and y-axis), to illustrate better the behaviour of the ATPi function.

We thank the reviewer for the suggestion. We have added Supplementary Figure 3, which presents a 3D visualisation of the ATP changes with respect to both resources and plasmid copy numbers. This figure enhances the manuscript by clearly illustrating the behaviour of the ATPi function across different conditions and contributes to a deeper understanding of the model.

- Figure 2 labels in the caption seem to be wrong.

The labels have been corrected in the new version of the Supplementary Information (Supp Info Lines 81-83).

- Regarding cell division, there are many strong theoretical works discussing and implementing different models for individual cellular division in bacteria, for example, standard adder/sizer/timer in cell physiology or the Cooper–Helmstetter model based on chromosome replication. Nevertheless, in this work, the division rule is based on ATP dynamics without any reference to previous works using same approach. Authors should discuss why a new model is proposed here when there are many alternatives in the literature.

We thank the reviewer for this insightful comment. We acknowledge the rich body of literature on bacterial cell division models, including the adder/sizer/timer and the Cooper–Helmstetter model. Our study, however, focuses on plasmid dynamics within bacterial cells, rather than on the intricacies of bacterial cell division per se. To this end, we employed an ATP-driven division rule as a proxy for energy availability and cellular health, which, while simpler than canonical cell division models, suffices for capturing the essential features of plasmid replication and segregation. We also use a standard Michaelis-Menten kinetics to model metabolic constraints under resource limitation.

We argue that while the choice of division model may slightly alter quantitative results, it does not qualitatively impact the central conclusions of our work concerning the modulation of the plasmid copy number distribution in response to selective pressures. Thus, our model offers a balance between simplicity and the ability to replicate the key dynamics necessary for the scope of our research. We hope this clarifies the rationale behind our methodological choices.

Page 4

- Usual growth bacterial curves are in \log_{10} (number of cells).

As recommended by the reviewer, we have modified Supplementary Figures 1 and 7 to show the y-axis in \log_{10} scale.

- Here for example lag-phase seems to be missing (although difficult to see without the log transformation), something that should be discussed, especially when simulating serial transfer experiments.

We acknowledge the importance of accurately representing the lag phase in serial transfer experiments, as it is a significant factor influencing population dynamics. We have updated Supplementary Figures 1 and 7 with a log-scale transformation for cell numbers, which has improved the visibility of the lag phase. This change has allowed us to identify and now discuss the variations in lag phase duration observed between different serial transfer episodes (Supp Info Lines 268-273).

- The concept of the maximal copy number and replication delay parameter seems to be included based on the author's expertise. Is there any mathematical model considering individual cell plasmid dynamics in the literature?

Thank you for raising this important point. There is a long tradition of mathematical models that focus on individual cell plasmid dynamics, such as the work by Paulsson in 2002 and others. These models often employ different approaches to model plasmid replication, for instance using a sigmoidal probability function. While different, our model is consistent with these studies in the assumption that the probability of replication is an asymptotically decreasing function of the existing intracellular plasmid concentration. In our manuscript, we refer to this asymptote as the 'maximum copy number' (Supp Info Lines 153-159)

The replication delay parameter in our model explicitly captures an essential feature of intracellular plasmid dynamics, namely that plasmids replicate at a faster time scale compared to the cell division process. We have included a paragraph (Supp Info Lines 160-170) describing how, in the absence of this parameter, cells may divide before plasmids have had a chance to replicate, thus resulting in

segregational instability (as shown in Supplementary Figure 8A). We acknowledge that further research is needed to understand the consequences of replication timing in different bacterial contexts, and we plan to investigate the implications of these variations in future work.

How the new model relates with population models of plasmid dynamics such as Fedorec, Alex JH, et al. "Two new plasmid post-segregational killing mechanisms for the implementation of synthetic gene networks in Escherichia coli." *Iscience* 14 (2019): 323-334?

Thank you for directing us to the study by Fedorec et al. (2019), which investigates plasmid stabilisation mechanisms in Escherichia coli, particularly focusing on post-segregational killing (PSK) systems. Their work provides a detailed exploration of genetic systems within plasmids, such as toxin-antitoxin systems and bacteriocins, and their role in enhancing plasmid persistence by affecting plasmid-free cells. Additionally, they develop mathematical models to describe the effects of these PSK mechanisms on plasmid stability. In contrast, our model emphasises the intrinsic dynamics of multicopy plasmid replication and segregation, along with the stochastic nature of these processes.

Our model's focus on natural variability in plasmid numbers and the impact of selective pressures could be seen as providing a foundational understanding upon which additional stabilising mechanisms, such as PSK, could act. It offers the potential for future extensions or combinations with the PSK approaches to explore how natural plasmid dynamics interact with engineered stabilisation strategies. In summary, while our model contributes to understanding the inherent dynamics of plasmid replication and segregation under various conditions, the work of Fedorec et al. complements this by detailing how engineered mechanisms can actively influence plasmid stability. This combination of perspectives could ultimately lead to a more comprehensive understanding of plasmid dynamics within bacterial populations.

Page 5

- Labels are missing in Fig. 5

Thank you for your observation regarding Figure 5. This figure presents a time-series of cell types from multiple traps/movies. We have included representative microscopy images from a replicate example at specific time-points, indicated by arrows. A scale bar is provided to denote size. Upon reviewing the figure, we have ensured that all necessary labels, including those for axes, legends, and units, are present for clarity. If there are additional labels that the reviewer feels are missing, we would greatly appreciate specific feedback to improve the figure further.

- Not clear how the parametrization was done, for example: which data is used for this? Which is the source for the 3 points used in the regression? is there not any more useful data in the literature? a linear regression with 3 points is too poor to infer conclusions.

We recognize the reviewer's concerns regarding the robustness of linear regression analyses and the preference for more data points to strengthen the inferential power. Indeed, the experimental manipulation of PCN presents significant challenges, which constrains the number of data points we can obtain. Our primary objective with the regression analysis, using the data points available, was to validate the positive slope of the relationship, thereby supporting the assumption of a linear correlation between gene dosage and drug resistance. This correlation is a crucial assumption of our model and is also substantiated by the data depicted in Figure 2D. Future experimental work aimed at gathering

more data points would be invaluable in refining this aspect of the model and represents an avenue for further research.

- A linear correlation between PCN and resistance is attributed to ref 6 on page 1 but not discussed here.

We thank the reviewer for pointing out the need for clarification regarding the correlation between PCN and resistance levels. While reference 6 on page 1 indeed demonstrates a correlation between PCN and protein expression levels, our model extends this concept by assuming a direct correlation between gene dosage, as reflected by PCN, and resistance levels. This assumption is grounded in the notion that higher numbers of resistance gene copies (due to higher PCN) would result in increased resistance phenotypes. Importantly, this assumption is supported by our own empirical data. In Figure 2D of our manuscript, we present evidence that illustrates a clear trend consistent with this assumption: as PCN increases, we observe a corresponding increase in resistance levels. This trend in our data reinforces the model's assumption and provides a real-world basis for the correlation between PCN and antibiotic resistance. Accordingly, we have corrected the Supplementary Information to accurately reflect this assumption (Supp Info Lines 32-33).

- IC was not defined in the text (inhibitory concentration?), again number of bacteria should be represented in log10 to allow comparison with usual time-kill curves (and see if there are tails or shoulders...)

We thank the reviewer for pointing out the oversight. We are now defining Inhibitory Concentration (IC) earlier in the Supplementary Information (Supp Info Lines 218-220). Additionally, to improve clarity and facilitate comparison with conventional dose-response curves, we have modified the y-axis of Supp Figure 6 (now Supp Figure 8) to display the number of bacteria on a log10 scale.

- More details on how the model mimics starvation could be interesting, is this starvation assumed before transfer to new media? depends on the number of cells or it is just decreasing the disponibility of ATP?

Thank you for inquiring about the implementation of starvation conditions within our model. Starvation is simulated by reducing the ATP levels of all cells selected for transfer to a new medium to 25% of their pre-transfer levels (Lines 268-273). This is a simplification meant to reflect the reduced energy state of cells that have exhausted nutrient supplies prior to the transfer. It does not directly depend on cell number but rather represents a uniform condition applied to all cells to simulate the metabolic constraints they would experience under nutrient depletion. This approach provides a consistent baseline for examining the impact of plasmid dynamics under resource-limited conditions.

- Mean PCN increases from 19 to 20 during exposure, Is only one plasmid on average so relevant in terms of population dynamics? See the next point related to this.

We appreciate the reviewer's focus on the change in mean PCN. It's important to note that while a single-plasmid increase in the mean may seem marginal, it is the shift in the distribution's tail that is critical in the context of antibiotic exposure. The antibiotics preferentially eliminate cells with fewer plasmids, leading to an increased mean as a secondary effect. The more relevant change is in the higher PCN range, where a small increase can significantly enhance resistance and affect population

dynamics. The precise shape of the PCN distribution is challenging to determine experimentally and theoretically; however, our computer simulations suggest that even subtle shifts can have profound impacts on the population's survival and proliferation in the presence of antibiotics.

- Fig. 7 growth curves should be in log10. This is the standard for many different reasons (additive noise, the many differences in order of magnitude...). The relevance is to see if there are changes in the orders of magnitude of the population, but not so relevant tiny changes for maximum numbers.

Following the reviewer's suggestion, we have now updated Supplementary Figure 8 to include bacterial counts on a log10 scale, adhering to the standard that facilitates the visualisation of orders of magnitude differences in population sizes. We also maintain the representation in absolute numbers alongside the log10 scale, as we believe it provides a comprehensive view that captures both the broad trends and the specific variations in population growth.

- It is not clear reading the text the relationship between cell division, ATP threshold and antibiotic action. Is this relationship related to the equation in page 5 such as $A/(m*\mu+b) > \text{antibiotic_action} + N(0,0.1)$? but the antibiotic concentration is already considered in this equation. I think that the text is not clear enough to infer the meaning and implementation of the parameter antibiotic_action. Note that the text states that the antibiotic action is related to ATP and cell division, but later states that shapes plasmid distribution (is this a consequence or the influence of the parameter in plasmid distribution is included explicitly in the model?).

Thanks for this observation. We have revised the Supplementary Information to more clearly articulate the relationship between cell division, ATP thresholds, and antibiotic action. In particular, we have elaborated on the text (Lines 285-287) to delineate how the 'antibiotic_action' parameter is integrated into the model. This affects both the availability of ATP for cell division and the subsequent impact on plasmid distribution.

- The replication delay concept is neither clear to me.

To succinctly explain, this parameter is incorporated into our model to more accurately reflect the observed biological phenomenon where plasmid replication can occur independently of the bacterial cell division cycle. It addresses the need to simulate a scenario where, even as cells prepare to divide, plasmids may continue to replicate, thus preventing an unrealistic synchronisation of plasmid and cell division rates. This is crucial for maintaining plasmid numbers through cell generations, which is a significant factor in the resulting plasmid population dynamics. We have illustrated the implications of this parameter in Supplementary Figure 5 and provided a more detailed explanation in the Supplementary Information (Lines 151-166).

- Not clear the change in dynamics at time 30, that induces the variability in plasmid distribution within the population.

This variability is primarily attributed to the initiation of cell division within the simulated population, which is influenced by two key factors outlined at the start of our simulations. Supplementary Figure 12 illustrates a case where no initial variation is introduced and therefore all cells begin with the same PCN (equal to the mean for the entire population) and identical ATP levels. However, as the simulation progresses and cells commence division around time 30, these starting conditions result in the

emergence of variability in PCN among the progeny due to the stochastic nature of plasmid segregation during cell division. We have elaborated on this process and its implications in the Supplementary Information (Supp Info Lines 324-328).

- Are the modelled dynamics of ATP equal for all the cases in Fig 9?

The core model parameters governing ATP dynamics are consistent across all simulations presented in Figure 9. However, the specific ATP trajectory of each cell within the simulations may differ due to interactions with varying plasmid copy numbers, which influence the metabolic load and consequently the ATP levels. We have ensured that these underlying principles are uniformly applied to all cases to facilitate direct comparisons of the outcomes under different plasmid distributions.

- There is a major issue here in the text stating that it was assumed normal distribution, but data shows a Gaussian trend. What is the difference between a normal distribution and a Gaussian trend? What is a Gaussian trend and which data shows this trend?

We acknowledge the reviewer's point for greater precision in terminology. In the revised manuscript, we have refined our language to ensure consistency and clarity (Supp Info Lines 340-344). We now use 'normal distribution' to describe the statistical pattern observed in our data, aligning with standard nomenclature.

- Which antibiotic dynamics are considered to go from the grey distribution to the red one?

The grey distribution represents the baseline PCN distribution within the bacterial population under drug-free conditions. When we introduce antibiotic selection pressure, it imposes a differential survival advantage on bacteria with certain PCN values. This selection pressure results in the red distribution, which reflects the reshaped PCN distribution that favours those bacteria with a higher number of plasmids (mean PCN of each case is annotated above each distribution). We have expanded the Supplementary Information for clarity (Supp Info Lines 345-350).

- It is interesting that for some distributions the mean of plasmids mean values increases with antibiotics, but not for others. If the antibiotic selects high plasmid copies, it is expected an increase in plasmid mean values for an environment with antibiotic pressure. The authors should clarify why this is not observed in the figures.

We agree with the reviewer's observation regarding the variability in the change of mean PCN values under antibiotic pressure. Our simulations indicate a general trend toward increasing mean PCN due to selection for higher plasmid counts, yet this trend is not consistent across all scenarios. Indeed, some distributions exhibit a pronounced increase in mean PCN, while others show a more modest rise. This variability can be attributed to several factors, including the starting distribution of PCN within the population, the specific selective pressures exerted by the antibiotic, and the intrinsic replication dynamics of the plasmids. Under certain conditions, the selection pressure may result in a significant shift in the mean PCN, while under different conditions, the shift may be comparatively minor. The complex interplay between the initial heterogeneity of the population and the dynamics of antibiotic selection shapes this response. We acknowledge that our current dataset does not allow us to dissect these dynamics completely and recognize it as a potential area for future investigation. We have included a discussion to this effect in the Supplementary Information (Supp Info Lines 351-355).

Reviewer #3 (Remarks to the Author):

The revised version 1 of the manuscript entitled "Plasmid-mediated phenotypic noise leads to transient antibiotic resistance in bacteria" now includes supplemental information that includes a detailed description of the computational model. I have one main comment regarding my previous comments (see 1.). Furthermore, I have to note that some changes need to be added (in the text and some figures), where the rebuttal letter indicates that a change has already been made (see remaining comments below).

We are thankful for the reviewer's recognition of the Supplementary Information. It is our intent to provide a comprehensive and transparent account of the computational model.

The coefficient of variation of the maximal PCN (σ) is a crucial computational model parameter. To my understanding of the study, this parameter σ and several other model parameters were set without experimental validation. It would be interesting to estimate the PCN number variation of pBGT from the GFP intensities to provide proof for the chosen parameter of 15% (10%, see minor comment on Table S2) for the computer simulations.

We appreciate the reviewer's attention to the importance of the coefficient of variation (CoV) of the PCN distribution in our computational model. We understand the concerns regarding the experimental validation of this and other model parameters. In response, we wish to clarify that for the simulations presented within the manuscript, we have employed a 25% CoV, which, although conservative, allows us to cautiously underestimate the variability within the population. We have included this parameter in Table S2.

Recognizing the significance of the reviewer's point regarding the critical nature of variance in PCN, we have incorporated a new Supplementary Figure 9 in our revised manuscript. This figure provides a numerical assessment of the effects of varying both the mean and variance in the PCN distribution. Consistent with the reviewer's expectations, we observe that the variance in PCN is indeed a crucial parameter. Specifically, our simulations demonstrate that populations with low PCN variability display a step-function-like behaviour in survival probability. In contrast, those with high variability exhibit a nearly constant survival probability. This observation substantiates the choice of our parameter, which strikes a balance between the two extremes.

1. I recognize that the authors included some numbers in brackets below each colored circle in Fig. 2D; however, I do not find any explanation for these values in the figure caption, which makes the given numbers somewhat useless. (What do the numbers below each circle mean, and if they show the COV, why are they so different from the COV in Fig. 2E?). It is intuitive that a higher CV pronounced shift in the mean PCN. My main issue is that I still cannot understand why the authors chose to work with a small CV of only 10% when experimental measurements show higher variation in the GFP intensity. If a pronounced shift with a higher CV overestimates the change in the copy number, which is why smaller CVs are chosen, it would be necessary to show results from "these [additional] measurements" or simulations.

Thank you for your observations regarding the numbers included below each coloured circle in Fig. 2D. We realise that the lack of explanation for these values in the figure caption has led to some confusion, and we apologise for this oversight. To clarify, these numbers represent the coefficient of variation (CoV) for each represented bacterial population's Plasmid Copy Number (PCN), as measured by GFP intensity. Regarding the choice of a smaller CoV of 25% in our computational model, this was a deliberate decision to conservatively estimate the variance within bacterial populations. We understand the reviewer's concern that our experimental measurements indicate a higher variation in GFP intensity. To address this, we have included additional simulations in Supplementary Figure 9 that demonstrate the effects of higher CoVs on population survival. These simulations indicate that while a higher CoV can lead to significant shifts in mean PCN, our choice of a smaller CoV for the model deliberately underestimates this variability. This conservative approach ensures that any observed correlation between PCN variability and antibiotic resistance is likely to be a lower bound estimate, suggesting that the true impact of PCN variability on resistance could be even more pronounced than our model predicts. We have updated the figure legend (Lines 210-211) and expanded this discussion in the main text (Lines 163-169).

Figure 1D has moved to the SI with no unit in the x-axis label.

We thank the reviewer for pointing out this omission. We have updated Supplementary Figure 6 to include the appropriate units on the x-axis.

L125: There is no black line.

No, I do not find this corrected (Figure 1E, the black line is maybe below the red shade).

We apologise for any confusion caused by the versioning of the text. To clarify, we have made the necessary amendments, and the text now accurately refers to a 'red solid line' rather than a black line (Line 127-128). This change aligns the text with the corresponding figure, ensuring that there is no discrepancy between the figure description and what is depicted. Thank you for bringing this to our attention.

No, I do not find further text in the revised manuscript that provides clarity on the correlation between GFP and PCN.

We acknowledge the importance of establishing a clear correlation between GFP intensity and Plasmid Copy Number (PCN) within our study. This correlation is substantiated by both our experimental observations and prior research and described in the main text (Lines 162-170). Specifically, a recent study using fluorescent-reporter-based measurement techniques has demonstrated a positive correlation between PCN and protein expression, supporting our approach [Shao2021]. Consistent with these findings, our experimental data also exhibit a strong correlation between PCN, as measured by quantitative PCR, and GFP intensity, quantified via fluorescence spectrophotometry.

To further validate this correlation within our experimental framework, we employed cell sorting to categorise a population of MG/pBGT cells by GFP intensity into low, medium, and high fluorescence clusters. Subsequent qPCR analysis of these sorted groups confirmed the anticipated positive relationship between fluorescence intensity and mean PCN (as detailed in Figure S1). Moreover, this relationship is visually represented in Figure 2D, which illustrates the correlation between PCN and GFP in our specific data set, thereby reinforcing the validity of using GFP as a proxy for PCN in our

experimental system. We have performed a linear regression on this data and updated Figure 2D to illustrate this correlation (Lines 210-211).

Figure 2C: Something is peculiar with the x-axis ranges of Figure 2C. This still needs to be corrected.

We have updated this figure to correct the x-axis tick labels. We thank the reviewer for .

Figure 2E: It should be indicated that there is a jump on the x-axis between 0 and positive values shown on a logarithmic scale. What do the dotted lines show? If it results from a simple non-linear regression, the value for zero should not be included.

Thank you for pointing out the issue with the presentation of the x-axis in Figure 2E. We have now corrected the axis to clearly indicate the discontinuity and the logarithmic scale used for positive values. To address your concern about the inclusion of the zero value, we have now performed a linear regression analysis excluding the zero dose, which should provide a more accurate representation of the dose-response relationship. The revised figure and its legend have been updated to reflect these changes (Lines 212-213).

The dotted line cannot result from a linear regression with the actual AB doses, as the linear regression would be a linear function of the form $COV(x)=a+bx$, where x is the actual drug concentration. The function plotted on a linear-log scale (logarithmic x-scale with base 2 excluding $x=0$) would have an exponential form $COV(x)=a+b*2^x$. In other words, the difference $COV(x=1/32)-COV(x=1/64)$ would be 2 times higher as $COV(x=1/64)-COV(x=1/128)$ and so on. See Figure S1, but note that b is positive in Fig S1.

We appreciate the reviewer's detailed analysis of the regression line in Figure 2E. There was indeed a lack of clarity in our presentation of the axis and the nature of the regression line. The dotted line was intended to represent the model's fit to the data points based on dose numbers rather than the actual logarithmic concentrations of the antibiotic. We acknowledge this was misleading. To resolve this, we have now revised Figure 2E to plot the data on a true \log_2 scale for the antibiotic concentrations, ensuring that the displayed regression accurately reflects the relationship between COV and the actual concentrations.

I need help finding Figure S16.

Upon review, we found no reference to Figure S16 in our submission. It seems there may be a miscommunication or typographical error. We welcome further clarification from the reviewer.

L345 "strength of selection [for lower PCN]": Please clarify that you are referring to selection for lower PCN.

We appreciate the reviewer's request for clarification on the section discussing the dynamics of plasmid segregation and replication post-antibiotic exposure. To elucidate, in our computational model, we observe that after surviving an antibiotic challenge, cells with higher PCN resume growth. As these cells proliferate, the inherent stochastic nature of plasmid segregation and replication allows for the entire spectrum of the PCN distribution to be potentially reestablished by any surviving cell.

However, due to the fitness cost associated with carrying a higher number of plasmids, when the selective pressure of the antibiotic is removed, there is an inherent selection against maintaining a high PCN. Consequently, lower-PCN cells gain a relative competitive advantage, leading to a decrease in the mean PCN of the population over time, returning it to pre-exposure levels. This observation is consistent across different levels of selective pressures, where the strength of selection—initially favouring high-PCN cells under antibiotic stress—shifts to favour lower-PCN cells once the antibiotic pressure is removed, due to the fitness costs associated with plasmid carriage. We have rephrased this section of the manuscript for clarity (Lines 109-112).

L594: How often do plasmids replicate per generation? I quickly looked in the code and saw a parameter set to 3. Is that the maximum number of replications per minute? If yes, it is still unclear why a maximum number of three replications per minute was chosen.

We appreciate the reviewer's inquiry into the replication parameter. The value of three, as designated in our model's code, reflects the maximal potential replications a plasmid can undergo per minute. We acknowledge that in reality, plasmid replication is subject to a myriad of influences that can affect its rate. To simulate these conditions, we implemented an asymptotic probability function governing replication. Given the model's design where a cell division cycle spans approximately 30 iterations, it is entirely plausible—and indeed observed—that not all cells reach their maximal plasmid copy number before division. This leads to the natural formation of plasmid-free cells over time, thereby influencing the average plasmid copy number within the population. This deliberate aspect of our model seeks to emulate the complexities and variabilities observed in biological plasmid replication, ensuring that our simulations reflect the stochastic nature of these processes as closely as possible. We are discussing this assumption in Lines 160-170 of the Supplementary Information.

The model description should also describe this iterative mechanism of plasmid replication. The 'upper limit on the number of replications per minute' is indeed a model parameter.

We thank the reviewer for the opportunity to clarify the iterative mechanism of plasmid replication within our model. Our approach simulates plasmid replication as a stochastic process occurring at discrete time intervals, with each iteration offering a potential opportunity for a plasmid to replicate, subject to the probabilistic framework governing the system.

The 'upper limit on the number of replications per minute,' as mentioned in the code, is indeed a critical parameter that encapsulates the maximal replication frequency within a given time frame. This parameter is deeply rooted in both empirical observations and a survey of the existing literature on plasmid replication rates. By introducing this upper limit, we aim to reflect the natural constraints and variability inherent in biological systems, as discussed in Lines 153-159 of the Supplementary Information.

Each iteration in the model assesses the probability of replication, which diminishes asymptotically as the number of plasmids approaches the cell's individual maximum capacity. This iterative process continues until the cell divides, with the potential for some plasmids to not replicate if the maximum replication rate is reached. This aspect of the model is consistent with the framework employed by earlier studies, such as the seminal work of Paulsson, while allowing for flexibility and adaptability to a

range of biological scenarios. It also permits us to account for the biological reality that cells can divide without all plasmids having replicated, contributing to the dynamic nature of plasmid populations.

Additional comments.

If one looks closely, the tip of the curve in Figure 1D is not visible anymore.

Thanks. We have corrected this figure.

REVIEWERS' COMMENTS

Reviewer #2 (Remarks to the Author):

I would like to thank the authors for their careful revision of the article and the supplementary information, as well as for their answers to all my comments which have been addressed satisfactorily. I think that the findings, integrating theory from agent-based modelling and experimental data, may be of relevance to a better understanding of antibiotic resistance, and I encourage the editor to accept this paper for publication.

I have just a few minor comments the authors might want to consider.

Regarding the linear relationship between PCN and resistance three comments and a final recommendation

- It is mentioned in the rebuttal letter to review #2 that to prove that the model holds true over a range of PCN averages the reviewer should look at Figure 9. I assume that the reference is to figure 9A (in 9B no change in PCN averages is done). However, Figure 9A shows a tautology: if a linear relationship between resistance and PCN is assumed in the model, that should be the result of the model simulation, for this a correlation coefficient of 1. That is not proof of the statement. I think it is best to remove the R coefficient of this figure and be careful with pointing to the figure as a proof of linearity between PCN and resistance.

- In other parts of the supp info authors again try to justify this assumption by referring to Figure 7 in supp info (Ampicillin concentration vs blaTEM copies). In this regard I also have major concerns, this is not a valid regression, there are only 2 experimental points and a third point based on knowledge (at 0,0), but one of the experimental points is just close to the origin. In other words, of course, it shows a linear trend, because in practical terms there are only 2 data points. I understand the difficulty of retrieving more experimental data, but therefore the linear trend is an assumption, and data only show in the best case a positive trend.

- The authors also point in the rebuttal letter to Figure 2D to demonstrate this linearity, but I do not see it (Figure 2D is PCN vs GFP intensity, or at least more details are needed if it is because of different populations have different resistance and therefore different PCN and GFP intensity).

Therefore, there seem to be only small clues indicating that a linear trend is possible, but it should be clearly stated that it is still an assumption. Any indication of Figure 2D as supporting evidence should be removed or explained better. I think it will help future works not to take this for granted, and instead conduct more research on this.

Regarding the log-scale authors mentioned that figures 1 and 7 in supp info were now in log-scale, I guess that they refer to other figures (4B and 10?) as fig. 1 and 7 do not show bacterial counts. In any case, this is solved as authors have changed (or added this scale) in relevant panels.

In line 86 of the manuscript it is written "assuming a linear relationship between PCN and gene dosage". I suggest changing the wording of the term "gene dosage"

Reviewer #3 (Remarks to the Author):

The comments on the previous version has been answered and the revised manuscript includes the consequent modifications.

A minor comment on the abstract: the formulation "multicopy plasmids, which are small DNA molecules" may be misleading here - most antibiotic resistance plasmids in Escherichia are rather large (in the context of plasmid size).

Reviewer #2 (Remarks to the Author):

I would like to thank the authors for their careful revision of the article and the supplementary information, as well as for their answers to all my comments which have been addressed satisfactorily. I think that the findings, integrating theory from agent-based modelling and experimental data, may be of relevance to a better understanding of antibiotic resistance, and I encourage the editor to accept this paper for publication.

We are very thankful for the positive feedback.

I have just a few minor comments the authors might want to consider.

Regarding the linear relationship between PCN and resistance three comments and a final recommendation

- It is mentioned in the rebuttal letter to review #2 that to prove that the model holds true over a range of PCN averages the reviewer should look at Figure 9. I assume that the reference is to figure 9A (in 9B no change in PCN averages is done). However, Figure 9A shows a tautology: if a linear relationship between resistance and PCN is assumed in the model, that should be the result of the model simulation, for this a correlation coefficient of 1. That is not proof of the statement. I think is best to remove the R coefficient of this figure and be careful with pointing to the figure as a proof of linearity between PCN and resistance.

Yes, we agree. We have modified the figure and figure caption (Supplementary Info Lines 204-209)

- In other parts of the supp info authors again try to justify this assumption by referring to Figure 7 in supp info (Ampicillin concentration vs blaTEM copies). In this regard I also have major concerns, this is not a valid regression, there are only 2 experimental points and a third point based on knowledge (at 0,0), but one of the experimental points is just close to the origin. In other words, of course, it shows a linear trend, because in practical terms there are only 2 data points. I understand the difficulty of retrieving more experimental data, but therefore the linear trend is an assumption, and data only show in the best case a positive trend.

We have changed this paragraph (Supplementary Info Lines 197-202)

- The authors also point in the rebuttal letter to Figure 2D to demonstrate this linearity, but I do not see it (Figure 2D is PCN vs GFP intensity, or at least more details are needed if it is because of different populations have different resistance and therefore different PCN and GFP intensity).

Therefore, there seem to be only small clues indicating that a linear trend is possible, but it should be clearly stated that it is still an assumption. Any indication of Figure 2D as supporting evidence should be removed or explained better. I think it will help future works not to take this for granted, and instead conduct more research on this.

We have removed this supplementary figure to avoid any misunderstandings.

Regarding the log-scale authors mentioned that figures 1 and 7 in supp info were now in log-scale, I guess that they refer to other figures (4B and 10?) as fig. 1 and 7 do not show bacterial counts. In any case, this is solved as authors have changed (or added this scale) in relevant panels.

OK.

In line 86 of the manuscript it is written "assuming a linear relationship between PCN and gene dosage". I suggest changing the wording of the term "gene dosage"

Thanks. We are now using the term "the expression level of plasmid-encoded genes."

Reviewer #3 (Remarks to the Author):

The comments on the previous version has been answered and the revised manuscript includes the consequent modifications.

A minor comment on the abstract: the formulation "multicopy plasmids, which are small DNA molecules" may be misleading here - most antibiotic resistance plasmids in Escherichia are rather large (in the context of plasmid size).

Thanks, we've rephrased the abstract.